# PROVABLE SIM-TO-REAL TRANSFER IN CONTINUOUS DOMAIN WITH PARTIAL OBSERVATIONS

**Jiachen Hu**[*]
School of Computer Science, Peking University
NickH@pku.edu.cn

**Han Zhong**[*]
Center for Data Science, Peking University
hanzhong@stu.pku.edu.cn

**Chi Jin**
Department of Electrical and Computer Engineering, Princeton University
chij@princeton.edu

**Liwei Wang**
National Key Laboratory of General Artificial Intelligence,
School of Intelligence Science and Technology, Peking University
Center for Data Science, Peking University, Beijing Institute of Big Data Research
wanglw@cis.pku.edu.cn

## ABSTRACT

Sim-to-real transfer, which trains RL agents in the simulated environments and then deploys them in the real world, has been widely used to overcome the limitations of gathering samples in the real world. Despite the empirical success of the sim-to-real transfer, its theoretical foundation is much less understood. In this paper, we study the sim-to-real transfer in *continuous* domain with *partial observations*, where the simulated environments and real-world environments are modeled by linear quadratic Gaussian (LQG) systems. We show that a popular robust adversarial training algorithm is capable of learning a policy from the simulated environment that is competitive to the optimal policy in the real-world environment. To achieve our results, we design a new algorithm for infinite-horizon average-cost LQGs and establish a regret bound that depends on the intrinsic complexity of the model class. Our algorithm crucially relies on a novel *history clipping* scheme, which might be of independent interest.

## 1 INTRODUCTION

Deep reinforcement learning has achieved great empirical successes in various real-world decision-making problems, such as Atari games (Mnih et al., 2015), Go (Silver et al., 2016; 2017), and robotics control (Kober et al., 2013). In addition to the power of large-scale deep neural networks, these successes also critically rely on the availability of a tremendous amount of data for training. For these applications, we have access to efficient simulators, which are capable of generating millions to billions of samples in a short time. However, in many other applications such as auto-driving (Pan et al., 2017) and healthcare (Wang et al., 2018), interacting with the environment repeatedly and collecting a large amount of data is costly and risky or even impossible.

A promising approach to solving the problem of data scarcity is *sim-to-real transfer* (Kober et al., 2013; Sadeghi & Levine, 2016; Tan et al., 2018; Zhao et al., 2020), which uses simulated environments to generate simulated data. These simulated data is used to train the RL agents, which will be then deployed in the real world. These trained RL agents, however, may perform poorly in real-world environments owing to the mismatch between simulation and real-world environments. This mismatch is commonly referred to as the *sim-to-real gap*.

To close such a gap, researchers propose various methods including (1) system identification (Kristinsson & Dumont, 1992), which builds a precise mathematical model for the real-world envi-

---

[*]Equal Contribution.

ronment; (2) domain randomization (Tobin et al., 2017), which randomizes the simulation and trains an agent that performs well on those randomized simulated environments; and (3) robust adversarial training (Pinto et al., 2017b), which finds a policy that performs well in a bad or even adversarial environment. Despite their empirical successes, these methods have very limited theoretical guarantees. A recent work Chen et al. (2021) studies the domain randomization algorithms for sim-to-real transfer, but this work has two limitations: First, their results (Chen et al., 2021, Theorem 4) heavily rely on domain randomization being able to sample simulated models that are very close to the real-world model with at least constant probability, which is hardly the case for applications with continuous domain. Second, their theoretical framework do not capture the problem with partial observations. However, sim-to-real transfer in continuous domain with partial observations is very common. Take dexterous in-hand manipulation (OpenAI et al., 2018) as an example, the domain consists of angels of different joints which is continuous and the training on the simulator has partial information due to the observation noises (for example, the image generated by the agent's camera can be affected by the surrounding environment). Therefore, we ask the following question:

*Can we provide a rigorous theoretical analysis for the sim-to-real gap in continuous domain with partial observations?*

This paper answers the above question affirmatively. We study the sim-to-real gap of the robust adversarial training algorithm and address the aforementioned limitations. To summarize, our contributions are three-fold:

- We use finite horizon LQGs to model the simulated and real-world environments, and also formalize the problem of sim-to-real transfer in *continuous* domain with *partial observations*. Under this framework, the learner is assumed to have access to a simulator class $\mathscr{E}$, each of which represents a simulator with certain control parameters. We analyze the sim-to-real gap (Eqn. (6)) of the *robust adversarial training* algorithm trained in simulator class $\mathscr{E}$. Our results show that the sim-to-real gap of the robust adversarial training algorithm is $\tilde{O}(\sqrt{\delta_{\mathscr{E}} H})$. Here $H$ is the horizon length of the real task, and $\delta_{\mathscr{E}}$ denotes some intrinsic complexity of the simulator class $\mathscr{E}$. This result shows that the sim-to-real gap is small for simple simulator classes and short tasks, while it gets larger for complicated classes and long tasks. By establishing a nearly matching lower bound, we further show this sim-to-real gap enjoys a near-optimal rate in terms of $H$.

- To bound the sim-to-real gap of the robust adversarial training algorithm, we develop a new reduction scheme that reduces the problem of bounding the sim-to-real gap to designing a sample-efficient algorithm in infinite-horizon average-cost LQGs. Our reduction scheme for LQGs is different from the one in Chen et al. (2021) for MDPs because the value function (or more precisely, the optimal bias function) is not naturally bounded in a LQG instance, which requires more sophisticated analysis.

- To prove our results, we propose a new algorithm, namely LQG-VTR, for infinite-horizon average-cost LQGs with *convex cost functions*. Theoretically, we establish a regret bound $\tilde{O}(\sqrt{\delta_{\mathscr{E}} T})$ for LQG-VTR, where $T$ is the number of steps. To the best of our knowledge, this is the first "instance-dependent" result that depends on the intrinsic complexity of the LQG model class with convex cost functions, whereas previous works only provide worst-case regret bounds depending on the ambient dimensions (i.e., dimensions of states, controls, and observations).

At the core of LQG-VTR is a *history clipping* scheme, which uses a clipped history instead of the full history to estimate the model and make predictions. This history clipping scheme helps us reduce the intrinsic complexity $\delta_{\mathscr{E}}$ exponentially from $O(T)$ to $O(\text{poly}(\log T))$ (cf. Appendix G).

Our theoretical results also have two implications: First, the robust training algorithm is provably efficient (cf. Theorem 1) in continuous domain with partial observations. One can turn to the robust adversarial training algorithm if the domain randomization has poor performance (cf. Appendix A.4). Second, for stable LQG systems, only a short clipped history needs to be remembered to make accurate predictions in the infinite-horizon average-cost setting.

## 1.1 RELATED WORK

**Sim-to-real Transfer**   Sim-to-real transfer, using simulated environments to train a policy that can be transferred to the real world, is widely used in many realistic scenarios such as robotics (e.g., Rusu et al., 2017; Tan et al., 2018; Peng et al., 2018; OpenAI et al., 2018; Zhao et al., 2020). To close the sim-to-real gap, various empirical algorithms are proposed, including robust adversarial training (Pinto et al., 2017b), domain adaptation (Tzeng et al., 2015), inverse dynamics methods (Christiano et al., 2016), progressive networks (Rusu et al., 2017), and domain randomization (Tobin et al., 2017). In this work, we focus on the robust adversarial training algorithm. Jiang (2018); Feng et al. (2019); Zhong et al. (2019) studies the sim-to-real transfer theoretically, but they require real-world samples to improve the policy during the training phase, while our work does not use any real-world samples. Our work is mostly related to Chen et al. (2021), which studies the benefits of domain randomization for sim-to-real transfer. As mentioned before, however, Chen et al. (2021) cannot tackle the sim-to-real transfer in continuous domain with partial observations, which is the focus of our work.

**Robust Adversarial Training Algorithm**   There are many empirical works studying robust adversarial training algorithms. Pinto et al. (2017b) proposes the robust adversarial training algorithm, which trains a policy in the adversarial environment. Then Pinto et al. (2017a); Mandlekar et al. (2017); Pattanaik et al. (2017); Dennis et al. (2020) show that the robust policy obtained by the robust adversarial training method can achieve good performance in the real world. But these works lack theoretical guarantees. Broadly speaking, the robust adversarial training method is also related to the min-max optimal control (Ma et al., 1999; Ma & Braatz, 2001) and robust RL (Morimoto & Doya, 2005; Iyengar, 2005; Xu & Mannor, 2010; Ho et al., 2018; Tessler et al., 2019; Mankowitz et al., 2019; Goyal & Grand-Clement, 2022).

**LQR and LQG**   There is a line of works (Mania et al., 2019; Cohen et al., 2019; Simchowitz & Foster, 2020; Lale et al., 2020a; Chen & Hazan, 2021; Lale et al., 2022) studying the infinite-horizon linear quadratic regulator (LQR), where the learner can observe the state. For the more challenging infinite-horizon LQG control, where the learner can only observe the noisy observations generated from the hidden state, Mania et al. (2019); Simchowitz et al. (2020); Lale et al. (2020c; 2021) propose various algorithms and establish regret bound for them. In specific, Mania et al. (2019); Lale et al. (2020b) study the *strongly convex* cost setting, where it is possible to achieve $O(\text{poly}(\log T))$ regret bound. Simchowitz et al. (2020); Lale et al. (2020c; 2021) and our work focus on the *convex* cost, where Simchowitz et al. (2020); Lale et al. (2020c) establish a $T^{2/3}$ worst-case regret. Lale et al. (2021) derives a $T^{1/2}$ worst-case regret depending on the ambient dimensions, which is based on strong assumptions and complicated regret analysis. In contrast, with *weaker assumptions and cleaner analysis*, our results only depend on the *intrinsic complexity* of the model class, which might be potentially small (cf. Appendix G).

## 2 PRELIMINARIES

### 2.1 FINITE-HORIZON LINEAR QUADRATIC GAUSSIAN

We consider the following finite-horizon linear quadratic Gaussian (LQG) model:

$$x_{h+1} = Ax_h + Bu_h + w_h, \quad y_h = Cx_h + z_h, \tag{1}$$

where $x_h \in \mathbb{R}^n$ is the hidden state at step $h$; $u_h \in \mathbb{R}^m$ is the action at step $h$; $w_h \sim \mathcal{N}(\mathbf{0}, I_n)$ is the process noise at step $h$; $y_h \in \mathbb{R}^p$ is the observation at step $h$; and $z_h \sim \mathcal{N}(\mathbf{0}, I_p)$ is the measurement noise at step $h$. Here the noises are i.i.d. random vectors. The initial state $x_0$ is assumed to follow a Gaussian distribution. Moreover, we denote by $\Theta := (A \in \mathbb{R}^{n \times n}, B \in \mathbb{R}^{n \times m}, C \in \mathbb{R}^{p \times n})$ the parameters of this LQG problem.

The learner interacts with the system as follows. At each step $h$, the learner observes an observation $y_h$, chooses an action $u_h$, and suffers a loss $c_h = c(y_h, u_h)$, which is defined by

$$c(y_h, u_h) = y_h^\top Q y_h + u_h^\top R u_h,$$

where $Q \in \mathbb{R}^{p \times p}$ and $R \in \mathbb{R}^{m \times m}$ are known positive definite matrices.

For the finite-horizon setting, the interaction ends after receiving the cost $c_H$, where $H$ is a positive integer. Let $\mathcal{H}_h = \{y_0, u_0, \cdots, y_{h-1}, u_{h-1}, y_h\}$ be the history at step $h$. Given a policy $\pi = \{\pi_h : \mathcal{H}_h \to u_h\}_{h=0}^H$, its expected total cost is defined by

$$V^\pi(\Theta) = \mathbb{E}_\pi \Big[ \sum_{h=0}^H y_h^\top Q y_h + u_h^\top R u_h \Big],$$

where the expectation is taken with respect to the randomness induced by the underlying dynamics and policy $\pi$. The learner aims to find the optimal policy $\pi^\star$ with *minimal expected total cost*, which is defined by $\pi^\star = \arg\min_\pi V^\pi(\Theta)$. For simplicity, we use the notation $V^\star(\Theta) = V^{\pi^\star}(\Theta)$.

## 2.2 Infinite-horizon Linear Quadratic Gaussian

For the infinite-horizon average-cost LQG, we use the $t$ and $J$ to denote the time step and expected total cost function respectively to distinguish them from the finite-horizon setting. Similar to the finite-horizon setting, the learner aims to find a policy $\pi = \{\pi_t : \mathcal{H}_t \to u_t\}_{t=0}^\infty$ that minimizes the expected total cost $J^\pi(\Theta)$, which is defined by

$$J^\pi(\Theta) = \lim_{T\to\infty} \frac{1}{T} \mathbb{E}_\pi \Big[ \sum_{t=0}^T y_t^\top Q y_t + u_t^\top R u_t \Big].$$

The optimal policy $\pi_{\text{in}}^\star$ is defined by $\pi_{\text{in}}^\star \overset{\text{def}}{=} \arg\min_\pi J^\pi(\Theta)$. We also use the notation $J^\star(\Theta) = J^{\pi_{\text{in}}^\star}(\Theta)$. We measure the $T$-step optimality of the learner's policy $\pi$ by its regret:

$$\text{Regret}(\pi; T) = \sum_{t=0}^T \mathbb{E}_\pi[c_t - J^\star(\Theta)]. \tag{2}$$

Throughout this paper, when $\pi$ is clear from the context, we may omit $\pi$ from $\text{Regret}(\pi; T)$.

It is known that the optimal policy for this problem is a linear feedback control policy, i.e., $u_t = -K(\Theta)\hat{x}_{t|t,\Theta}$, where $K(\Theta)$ is the optimal control gain matrix and $\hat{x}_{t|t,\Theta}$ is the belief state at step $t$ (i.e. the estimated mean of $x_t$). One can use the Kalman filter (Kalman, 1960) to obtain $\hat{x}_{t|t,\Theta}$ and dynamic programming to obtain $K(\Theta)$ when the system $\Theta$ is known. In particular, denote $P(\Theta)$ as the unique solution to the discrete-time algebraic Riccati equation (DARE):

$$P(\Theta) = A^\top P(\Theta) A + C^\top Q C - A^\top P(\Theta) B (R + B^\top P(\Theta) B)^{-1} B^\top P(\Theta) A,$$

then $K(\Theta)$ can be obtained using $P(\Theta)$. We also use $\Sigma(\Theta)$ to denote the steady-state covariance matrix of $x_t$. More details are deferred to Appendix A.3.

The LQG instance (1) can be depicted in the predictor form (Kalman, 1960; Lale et al., 2020b; 2021)

$$x_{t+1} = (A - F(\Theta)C)x_t + Bu_t + F(\Theta)y_t, \quad y_t = Cx_t + e_t, \tag{3}$$

where $F(\Theta) = AL(\Theta)$ and $e_t$ denotes a zero-mean innovation process.

**Bellman Equation** We define the optimal bias function of $\Theta = (A, B, C)$ for $(\hat{x}_{t|t,\Theta}, y_t)$ as

$$h_\Theta^\star\left(\hat{x}_{t|t,\Theta}, y_t\right) \overset{\text{def}}{=} \hat{x}_{t|t,\Theta}^\top (P(\Theta) - C^\top Q C)\hat{x}_{t|t,\Theta} + y_t^\top Q y_t. \tag{4}$$

With this notation, the Bellman optimality equation (Lale et al., 2020c, Lemma 4.3) is given by

$$J^\star(\Theta) + h_\Theta^\star\left(\hat{x}_{t|t,\Theta}, y_t\right) = \min_u \left\{ c(y_t, u) + \mathbb{E}_{\Theta,u}\left[h_\Theta^\star\left(\hat{x}_{t+1|t+1,\Theta}, y_{t+1}\right)\right] \right\}, \tag{5}$$

where the equality is achieved by the optimal control of $\Theta$.

**Notation** We use $O(\cdot)$ notation to highlight the dependency on $H, n, m, p$, yet omit the polynomial dependency on some complicated instance-dependent constants ($\tilde{O}(\cdot)$ further omits polylogarithmic factors). For function $f : \mathcal{X} \to \mathbb{R}$, its $\ell_\infty$-norm $\|f\|_\infty$ is defined by $\sup_{x\in\mathcal{X}} f(x)$. We also define $\|\mathcal{F}\|_\infty \overset{\text{def}}{=} \sup_{f\in\mathcal{F}} \|f\|_\infty$. Let $\rho(\cdot)$ denote the spectral radius, i.e., the maximum absolute value of eigenvalues.

## 2.3 SIM-TO-REAL TRANSFER

The principal framework of sim-to-real transfer works as follows: the learner trains a policy in the simulators of the environment, and then applies the obtained policy to the real world. We follow Chen et al. (2021) to present a formulation of the sim-to-real transfer. The simulators are modeled as a set of finite-horizon LQGs with control parameters (e.g., physical parameters, control delays, etc.), where different parameters correspond to different dynamics. We denote this simulator class by $\mathscr{E}$. To make the problem tractable, we also impose the realizability assumption: the real-world environment $\Theta^\star \in \mathscr{E}$ is contained in the simulator set.

Now we describe the sim-to-real transfer paradigm formally. In the simulation phase, the learner is given the set $\mathscr{E}$, each of which is a parameterized simulator (LQG system). During the simulation phase, the learner can interact with each simulator for arbitrary times. However, the learner does NOT know which one represents the real-world environment, which may cause the learned policy to perform poorly in the real world. This challenge is commonly referred to as the *sim-to-real gap*. Mathematically, assuming the learned policy in the simulation phase is $\pi(\mathscr{E})$, its sim-to-real gap is defined by

$$\mathrm{Gap}(\pi(\mathscr{E})) = V^{\pi(\mathscr{E})}(\Theta^\star) - V^\star(\Theta^\star), \tag{6}$$

which is the difference between the cost of simulation policy $\pi(\mathscr{E})$ on the real-world model and the optimal cost in the real world.

## 2.4 ROBUST ADVERSARIAL TRAINING ALGORITHM

With our sim-to-real transfer framework defined above, we now formally define the robust adversarial training algorithm used in the simulator training procedure.

**Definition 1** (Robust Adversarial Training Oracle). *The robust adversarial training oracle returns a (history-dependent) policy $\pi_{\mathrm{RT}}$ such that*

$$\pi_{\mathrm{RT}} = \arg\min_\pi \max_{\Theta \in \mathscr{E}} [V^\pi(\Theta) - V^\star(\Theta)], \tag{7}$$

*where $V^\star$ is the optimal cost, and $V^\pi$ is the cost of $\pi$, both on the LQG model $\Theta$. Note that the real world model $\Theta^\star$ is unknown to the robust adversarial training oracle.*

This robust adversarial training oracle aims to find a policy that minimizes the worst case value gap. This oracle can be achieved by many algorithms in min-max optimal control (Ma et al., 1999; Ma & Braatz, 2001) and robust RL (Morimoto & Doya, 2005; Iyengar, 2005; Xu & Mannor, 2010; Pinto et al., 2017a;b; Ho et al., 2018; Tessler et al., 2019; Mankowitz et al., 2019; Kuang et al., 2022; Goyal & Grand-Clement, 2022).

## 3 MAIN RESULTS

Before presenting our results, we introduce several standard notations and assumptions for LQGs.

**Assumption 1.** *The real world LQG system $\Theta^\star = (A^\star, B^\star, C^\star)$ is **open-loop stable**, i.e., $\rho(A^\star) < 1$. Assume*

$$\Phi(A^\star) \overset{\mathrm{def}}{=} \sup_{\tau \geq 0} \frac{\|(A^\star)^\tau\|_2}{\rho(A^\star)^\tau} < +\infty.$$

*We also assume that $(A^\star, B^\star)$ and $(A^\star, F(\Theta^\star))$ are controllable, $(A^\star, C^\star)$ is observable.*

For completeness, we provide the definitions of controllable, observable, and stable systems in Appendix A.2. Assumption 1 is common in previous works studying the regret minimization or system identification of LQG (Lale et al., 2021; 2020b; Oymak & Ozay, 2019). The bounded $\Phi(A^\star)$ condition is a mild condition as noted in Lale et al. (2020c); Oymak & Ozay (2019); Mania et al. (2019), which is satisfied when $A^\star$ is diagonalizable. We emphasize that the $(A^\star, F(\Theta^\star))$-controllable assumption here only helps us simplify the analysis (can be removed by more sophisticated analysis). It is not a necessary condition for our results.

In the framework of sim-to-real transfer, we have access to a parameterized simulator class $\mathscr{E}$. It is natural to assume $\mathscr{E}$ has a bounded norm. The stability of the systems relies on the *contractible* property of the close-loop control matrix $A - BK$ (Lale et al., 2020c; 2021) . Thus, we make the following assumption, which is also used in Lale et al. (2020c; 2021).

**Assumption 2.** *The simulators* $\Theta' = (A', B', C') \in \mathscr{E}$ *have **contractible** close-loop control matrices:* $\|K(\Theta')\|_2 \leq N_K$ *and* $\|A' - B'K(\Theta')\|_2 \leq \gamma_1$ *for fixed constant* $N_K > 0, 0 < \gamma_1 < 1$. *We assume there exists constant* $N_S$ *such that for any* $\Theta' \in \mathscr{E}$, $\|A'\|_2, \|B'\|_2, \|C'\|_2 \leq N_S$.

As we study the partially observable setting with infinite-horizon average cost, we also require the belief states of any LQG model in $\mathscr{E}$ to be stable and convergent (Mania et al., 2019; Lale et al., 2021). Since the dynamics for belief states follow the predictor form in (3), we assume matrix $A - F(\Theta)C$ is stable as follows.

**Assumption 3.** *Assume there exists constant* $(\kappa_2, \gamma_2)$ *such that for any* $\Theta' = (A', B', C') \in \mathscr{E}$, *the matrix* $A' - F(\Theta')C'$ *is* $(\kappa_2, \gamma_2)$-***strongly stable**. *Here* $A' - F(\Theta')C'$ *is* $(\kappa_2, \gamma_2)$-*strongly stable means* $\|F(\Theta')\|_2 \leq \kappa_2$ *and there exists matrices* $L$ *and* $G$ *such that* $A' - F(\Theta')C' = GLG^{-1}$ *with* $\|L\|_2 \leq 1 - \gamma_2, \|G\|_2\|G^{-1}\|_2 \leq \kappa_2$.

We assume $x_0 \sim \mathcal{N}(\mathbf{0}, I)$ for simplicity. Note that the Kalman filter converges exponentially fast to its steady-state (Caines & Mayne, 1970; Chan et al., 1984), we omit the error brought by $x_0$ starting at covariance $I$ instead of the $\Sigma(\Theta^\star)$ (see e.g., Appendix G of Lale et al. (2021)) without loss of generality.

Now we are ready to state our main theorem, which establishes a sharp upper bound for the sim-to-real gap of $\pi_{\mathrm{RT}}$.

**Theorem 1.** *Under Assumptions 1, 2, and 3, the sim-to-real gap of $\pi_{\mathrm{RT}}$ satisfies*

$$\mathrm{Gap}(\pi_{\mathrm{RT}}) \leq \tilde{O}\left(\sqrt{\delta_{\mathscr{E}} H}\right),$$

*where $\delta_{\mathscr{E}}$ (defined in Theorem 4) characterizes the complexity of model class $\mathscr{E}$.*

Here we use $\tilde{O}$ to highlight the dependency with respect to the dimensions of the problem (i.e., $m, n, p, H$) and hide the uniform constants, log factors, and complicated instance-dependent constants. We present the high-level ideas of our proof in Section 4. The full proof is deferred to Appendix E. The following proposition shows that $\delta_{\mathscr{E}}$ has at most logarithmic dependency on $H$.

**Proposition 1.** *Under Assumptions 1, 2, and 3, the intrinsic complexity $\delta_{\mathscr{E}}$ is always upper bounded by*

$$\delta_{\mathscr{E}} = \tilde{O}\left(\mathrm{poly}(m, n, p)\right).$$

The proof is deferred to Appendix G.1. Note that the sim-to-real gap of any trivial stable control policy is as large as $O(H)$ since stable policies incur at most $O(1)$ loss (compared with the optimal policy) at each step. Therefore, Theorem 1 along with proposition 1 ensures that $\pi_{\mathrm{RT}}$ is a highly non-trivial policy which only suffers $O(H^{-1/2})$ loss per step. Moreover, we can show that $\delta_{\mathscr{E}}$ will be smaller if $\mathscr{E}$ has more properties such as the low-rank structure. See Appendicies G.2 and G.3 for details.

To demonstrate the optimality of the upper bound in Theorem 1, we establish the following lower bound, which shows that the $\sqrt{H}$ sim-to-real gap is unavoidable. The proof is given in Appendix H.

**Theorem 2** (Lower Bound). *Under Assumptions 1, 2, and 3, for any history-dependent policy $\hat{\pi}$, there exists a model class $\mathscr{E}$ and a choice of $\Theta^\star \in \mathscr{E}$ such that :*

$$\mathrm{Gap}(\hat{\pi}) \geq \Omega(\sqrt{H}).$$

## 4 ANALYSIS

We provide a sketched analysis of the sim-to-real gap of $\pi_{\mathrm{RT}}$ in this section. In a nutshell, we first perform a reduction from bounding the sim-to-real gap to an infinite-horizon regret minimization problem. We further show that there exists a history-dependent policy achieving low regret bound

in the infinite-horizon LQG problem. This immediately implies that the sim-to-real gap of $\pi_{\text{RT}}$ will be small by reduction. Before coming to the analysis, we note first that any simulator $\Theta \in \mathscr{E}$ not satisfying Assumption 1 cannot be the real world system. Therefore, we can prune the simulator set $\mathscr{E}$ to remove the models that do not satisfy Assumption 1.

## 4.1 THE REDUCTION

Now we introduce the reduction technique, which connects the sim-to-real gap defined in Eqn. (6) and the regret defined in Eqn. (2).

**Lemma 3** (Reduction). *Under Assumptions 1, 2, and 3, the sim-to-real gap of $\pi_{\text{RT}}$ can be bounded by the $H$-step regret bound of any (history-dependent) policy $\pi$ defined in Eqn. (2):*

$$\text{Gap}(\pi_{\text{RT}}) \leq \text{Regret}(\pi; H) + D_h,$$

*where $D_h$ does not depend on $H$.*

Although the idea of reduction is also used in Chen et al. (2021), our reduction here is very different from theirs. Their proof relies on the communication assumption, which ensures that the optimal bias function in infinite-horizon MDPs is uniformly bounded. In contrast, the optimal bias function defined in (4) is not naturally bounded, which requires much more complicated analysis. See Appendix D for a detailed proof.

By Lemma 3, it suffices to construct a history-dependent policy $\hat{\pi}$ that has low regret. Since we can treat any history-dependent policy $\pi$ as an algorithm, it suffices to design an efficient regret minimization algorithm for the infinite-horizon average-cost LQG systems as $\hat{\pi}$.

## 4.2 THE REGRET MINIMIZATION ALGORITHM

Motivated by the reduction in the previous subsection, we construct $\hat{\pi}$ as a sample-efficient algorithm LQG-VTR (Algorithm 1). The key steps of LQG-VTR are summarized below.

**Model Selection** It has been observed by many previous works (e.g., Simchowitz & Foster (2020); Lale et al. (2021; 2020c); Tsiamis et al. (2020)) that an inaccurate estimation of the system leads to the actions that cause the belief state to explode (i.e., $\|\hat{x}_{t|t,\tilde{\Theta}}\|_2$ becomes linear or even super linear as $t$ grows). To alleviate this problem, we utilize a model selection procedure before the optimistic planning algorithm to stabilize the system. The model selection procedure rules out some unlikely systems from the simulator set, which ensures that the inaccuracies do not blow up during the execution of LQG-VTR. To this end, we collect a few samples with random actions, and use a modified system identification algorithm to estimate the system parameters (Lale et al., 2021; Oymak & Ozay, 2019). After the model selection procedure, we show that with high probability the belief states and observations stay bounded throughout the remaining steps (cf. Lemma 5 in Appendix C), so LQG-VTR does not terminate at Line 9.

**Estimate the Model with Clipped History** As the simulator class $\mathscr{E}$ is known to the agent, we use the value-target model regression procedure (Ayoub et al., 2020) to estimate the real-world model $\Theta^\star$ at the end of each episode $k$. To be more concrete, suppose the agent has access to the regression dataset $\mathcal{Z} = \{E_t\}_{t=1}^{|\mathcal{Z}|}$ at the end of episode $k$, where $E_t$ is $t$-th sample containing the belief state $\hat{x}_{t|t}$, action $u_t$, observation $y_t$, the estimated bias function $h_{\tilde{\Theta}}^\star$, and the regression target $(\hat{x}_{t+1|t+1}, y_{t+1})$. Here $\tilde{\Theta}$ is the optimistic model used in the $t$-th sample. Then, inspired by the Bellman equation in (5) we can estimate the model by minimizing the following least-squares loss

$$\hat{\Theta}'_{k+1} = \arg\min_{\Theta \in \mathcal{U}_1} \sum_{E_t \in \mathcal{Z}} \left( \mathbb{E}_{\Theta, u_t} \left[ h_{\tilde{\Theta}}^\star \left( \hat{x}'_{t+1|t+1}, y'_{t+1} \right) \mid \hat{x}_{t|t} \right] - h_{\tilde{\Theta}}^\star \left( \hat{x}_{t+1|t+1}, y_{t+1} \right) \right)^2, \quad (8)$$

where $\hat{x}'_{t+1|t+1}, y'_{t+1}$ denotes the random belief state and observation at step $t + 1$. However, it requires the full history at step $t$ to compute the expectation in Eqn. (8) as we show in Section 4.4, which leads to an $O(H)$ intrinsic complexity (i.e., $\delta_{\mathscr{E}} = O(H)$). Then the sim-to-real gap of $\pi_{\text{RT}}$ in Theorem 1 becomes $O(H)$, which is vacuous. Fortunately, we can use a *clipped* history (Line 12 of Algorithm 1) to compute an *approximation* of the expectation. Let $f_\Theta(\hat{x}_{t|t}, u, \tilde{\Theta}) \overset{\text{def}}{\approx}$

---

**Algorithm 1** LQG-VTR

---

1: Initialize: set model selection period length $T_w$ by Eqn. (56).
2:      set maximum state allowed $M_x = \bar{X}_1$ ($\bar{X}_1$ is defined in Lemma 5).
3:      set maximum number of episode $\bar{\mathcal{K}}$ by Eqn. (77).
4:      set $\psi$ by Lemma 9, $\beta$ by Eqn. (119), $l = O(\log(Hn + Hp))$.
5:      set $\mathcal{Z} = \mathcal{Z}_{new} = \emptyset$, initial state $x_0 \sim \mathcal{N}(\mathbf{0}, I)$, episode $k = 1$.
6: Compute $\mathcal{U}_1 = \text{Model Selection}(T_w, \mathscr{E})$ (Algorithm 2), $\tilde{\Theta}_1 = \arg\min_{\Theta \in \mathcal{U}_1} J^\star(\Theta)$.
7: **for** step $t = T_w + 1, \cdots, H$ **do**
8:   **if** $\|\hat{x}_{t|t,\tilde{\Theta}_k}\|_2 > M_x$ **then**
9:    Take $u_t = \mathbf{0}$ for the remaining steps and halt the algorithm.
10:   Compute the optimal action under system $\tilde{\Theta}_k$: $u_t = -K(\tilde{\Theta}_k)\hat{x}_{t|t,\tilde{\Theta}_k}$.
11:   Take action $u_t$ and observe $y_{t+1}$.
12:   Let $l_{clip} \stackrel{\text{def}}{=} \min(l, t)$, define $\tau_t \stackrel{\text{def}}{=} (y_t, u_{t-1}, y_{t-1}, u_{t-2}, y_{t-2}, ..., y_{t-l_{clip}+1}, u_{t-l_{clip}})$.
13:   Add sample $E_t \stackrel{\text{def}}{=} (\tau_t, \hat{x}_{t|t,\tilde{\Theta}_k}, \tilde{\Theta}_k, u_t, \hat{x}_{t+1|t+1,\tilde{\Theta}_k}, y_{t+1})$ to the set $\mathcal{Z}_{new}$.
14:   **if** importance score $\sup_{f_1, f_2 \in \mathcal{F}} \frac{\|f_1 - f_2\|_{\mathcal{Z}_{new}}^2}{\|f_1 - f_2\|_{\mathcal{Z}}^2 + \psi} \geq 1$ and $k < \bar{\mathcal{K}}$ **then**
15:    Add the history data $\mathcal{Z}_{new}$ to the set $\mathcal{Z}$, and reset $\mathcal{Z}_{new} = \emptyset$.
16:    Calculate $\hat{\Theta}_{k+1}$ using Eqn. (9).
17:    Update the confidence set $\mathcal{U}_{k+1} = \mathcal{C}(\hat{\Theta}_{k+1}, \mathcal{Z})$ by Eqn. (10).
18:    Compute $\tilde{\Theta}_{k+1} = \arg\min_{\Theta \in \mathcal{U}_{k+1}} J^\star(\Theta)$; episode counter $k = k + 1$.

---

$\mathbb{E}_{\Theta,u}[h_{\tilde{\Theta}}^\star(\hat{x}'_{t+1|t+1}, y'_{t+1}) \mid \hat{x}_{t|t}]$ be the approximation of the expectation (see Section 4.4 for the formal definition), we use $f$ to estimate $\Theta^\star$ with dataset $\mathcal{Z}$:

$$\hat{\Theta}_{k+1} = \arg\min_{\Theta \in \mathcal{U}_1} \sum_{E_t \in \mathcal{Z}} \left( f_\Theta\left(\hat{x}_{t|t}, u_t, \tilde{\Theta}\right) - h_{\tilde{\Theta}}^\star\left(\hat{x}_{t+1|t+1}, y_{t+1}\right) \right)^2. \tag{9}$$

With this estimator, we can construct the following confidence set,

$$\mathcal{C}\left(\hat{\Theta}_{k+1}, \mathcal{Z}\right) = \left\{ \Theta \in \mathcal{U}_1 : \sum_{E_t \in \mathcal{Z}} \left( f_\Theta(\hat{x}_{t|t}, u_t, \tilde{\Theta}) - f_{\hat{\Theta}_{k+1}}(\hat{x}_{t|t}, u_t, \tilde{\Theta}) \right)^2 \leq \beta \right\}, \tag{10}$$

**Update with Low-switching Cost**   Since the regret bound of LQG-VTR has a linear dependency on the number of times the algorithm switches the control policies (cf. Appendix E.5), we have to ensure the low-switching property (Auer et al., 2008; Bai et al., 2019) of our algorithm. We follow the idea of Kong et al. (2021); Chen et al. (2021) that maintains two datasets $\mathcal{Z}$ and $\mathcal{Z}_{new}$, representing current data used in model regression and new incoming data. The importance score (Line 14 of Algorithm 1) measures the importance of the data in $\mathcal{Z}_{new}$ with respect to the data in $\mathcal{Z}$. We only synchronize the current dataset with the new dataset and update the current policy when this value is greater than 1.

## 4.3 REGRET BOUND OF LQG-VTR

**Theorem 4.** *Under Assumptions 1, 2, and 3, the regret (Eq.(2)) of LQG-VTR is bounded by*

$$\tilde{O}\left(\sqrt{\delta_{\mathscr{E}} H}\right), \tag{11}$$

*where the intrinsic model complexity $\delta_{\mathscr{E}}$ is defined as*

$$\delta_{\mathscr{E}} \stackrel{\text{def}}{=} \dim_E(\mathcal{F}, 1/H) \log(\mathcal{N}(\mathcal{F}, 1/H))\|\mathcal{F}\|_\infty^2. \tag{12}$$

*Here we use function class $\mathcal{F}$ to capture the complexity of $\mathscr{E}$, which is defined in Section 4.4. $\dim_E(\mathcal{F}, 1/H)$ denotes the $1/H$-Eluder dimension of $\mathcal{F}$, $\mathcal{N}(\mathcal{F}, 1/H)$ is the $1/H$-covering number of $\mathcal{F}$. The definitions of Eluder dimension and covering number are deferred to Appendix A.2.*

Under Assumptions 1, 2, and 3, we obtain a $\sqrt{H}$-regret as desired because $\delta_{\mathscr{E}}$ is at most $\tilde{O}(\text{poly}(m, n, p))$ (cf. Appendix G). Notably, if we do not adopt the history clipping technique,

$\delta_{\mathscr{E}}$ will be linear in $H$, resulting in the regret bound becoming vacuous. Compared with existing works (Simchowitz et al., 2020; Lale et al., 2020c; 2021), we achieve a $\sqrt{H}$-regret bound for general model class $\mathscr{E}$ (depends on the intrinsic complexity of $\mathscr{E}$) with weaker assumptions and cleaner analysis. Theorem 1 is directly implied by Theorem 4 and the reduction in (75).

### 4.4 Construction of the Function Class $\mathcal{F}$

In this subsection, we present the intuition and construction of $\mathcal{F}$. To begin with, recall that the optimal bias function $h^\star_{\tilde{\Theta}}$ of the optimistic system $\tilde{\Theta}$ is

$$h^\star_{\tilde{\Theta}}\left(\hat{x}_{t+1|t+1,\tilde{\Theta}}, y_{t+1}\right) = \hat{x}^\top_{t+1|t+1,\tilde{\Theta}}\left(\tilde{P} - \tilde{C}^\top Q\tilde{C}\right)\hat{x}_{t+1|t+1,\tilde{\Theta}} + y^\top_{t+1}Qy_{t+1},$$

where $(\tilde{P}, \tilde{C})$ are the parameters with respect to $\tilde{\Theta}$. Given any underlying system $\Theta$, we know the next step observation $y_{t+1,\Theta}$ given belief state $\hat{x}_{t|t,\Theta}$ and action $u$ is defined as

$$y_{t+1,\Theta} = \mathcal{Y}\left(\hat{x}_{t|t,\Theta}, u, \Theta\right) \overset{\text{def}}{=} CA\hat{x}_{t|t,\Theta} + CBu + Cw_t + CA(x_t - \hat{x}_{t|t,\Theta}) + z_{t+1}.$$

Here we use $\mathcal{Y}$ to denote the stochastic process that generates $y_{t+1,\Theta}$ under system $\Theta$.

Thus, we can define function $f'$ to be the one-step Bellman backup of the optimal bias function $h^\star_{\tilde{\Theta}}$ *under the transition of the underlying system $\Theta$*:

$$f'\left(\tau^t, \hat{x}_{t|t,\tilde{\Theta}}, u, \tilde{\Theta}\right) \overset{\text{def}}{=} \mathbb{E}\left[h^\star_{\tilde{\Theta}}\left(\hat{x}_{t+1|t+1,\tilde{\Theta}}, y_{t+1,\Theta}\right)\right], \quad y_{t+1,\Theta} = \mathcal{Y}\left(\hat{x}_{t|t,\Theta}, u, \Theta\right). \tag{13}$$

The next step belief state $\hat{x}_{t+1|t+1,\tilde{\Theta}}$ is determined by the Kalman filter as

$$\hat{x}_{t+1|t+1,\tilde{\Theta}} = \left(I - \tilde{L}\tilde{C}\right)\left(\tilde{A}\hat{x}_{t|t,\tilde{\Theta}} + \tilde{B}u\right) + \tilde{L}y_{t+1,\Theta}. \tag{14}$$

However, we need the full history $\tau^t$ to compute $\hat{x}_{t|t,\Theta}$ as shown below, which is unacceptable. To mitigate this problem, we compute the *approximate belief state* $\hat{x}^c_{t|t,\Theta}$ with clipped length-$l$ history $\tau^l = \{u_{t-l}, y_{t-l+1}, \cdots, u_{t-1}, y_t\}$:

$$\hat{x}_{t|t,\Theta} = (I - LC)A\hat{x}_{t-1|t-1,\Theta} + (I - LC)Bu_{t-1} + Ly_t \quad \text{(require full history } \tau^t \text{ to compute } \hat{x}_{t|t,\Theta})$$

$$= (A - LCA)^l \hat{x}_{t-l|t-l,\Theta} + \underbrace{\sum_{s=1}^{l}(A - LCA)^{s-1}\left((I - LC)Bu_{t-s} + Ly_{t-s+1}\right)}_{\hat{x}^c_{t|t,\Theta}}. \tag{15}$$

Thanks to Assumption 3, we can show that $(A - LCA)^l \hat{x}_{t-l|t-l,\Theta}$ is a small term whose $\ell_2$ norm is bounded by $O(\kappa_2(1 - \gamma_2)^l)$ since $\|(I - LC)A\|_2 = \|A(I - LC)\|_2$. Therefore, $\hat{x}^c_{t|t,\Theta}$ will be a good approximation of $\hat{x}_{t|t,\Theta}$, helping us control the error of clipping the history.

With the above observation, we formally define $\mathcal{F}$, which is an approximation by replacing the full history with the clipped history. Let the domain of $\mathcal{F}$ be $\mathcal{X} \overset{\text{def}}{=} \left(\mathcal{B}^m_{\bar{U}} \times \mathcal{B}^p_{\bar{Y}}\right)^l \times \mathcal{B}^n_{\bar{X}_1} \times \mathcal{B}^m_{\bar{U}} \times \mathscr{E}$, where $\mathcal{B}^d_v = \{x \in \mathbb{R}^d : \|x\|_2 \leq v\}$ for any $(d, v) \in \mathbb{N} \times \mathbb{R}$. Here $\bar{U}, \bar{Y}$ and $\bar{X}_1$ will be specified in Lemma 5. We define $\mathcal{F}$ formally as follows:

$$\mathcal{F} \overset{\text{def}}{=} \{f_\Theta : \mathcal{X} \to \mathbb{R} \mid \Theta \in \mathscr{E}\},$$

$$f_\Theta\left(\tau^l, \hat{x}_{t|t,\tilde{\Theta}}, u, \tilde{\Theta}\right) \overset{\text{def}}{=} \mathbb{E}\left[h^\star_{\tilde{\Theta}}\left(\hat{x}_{t+1|t+1,\tilde{\Theta}}, y^c_{t+1,\Theta}\right)\right], \quad y^c_{t+1,\Theta} = \mathcal{Y}\left(\hat{x}^c_{t|t,\Theta}, u, \Theta\right). \tag{16}$$

Here $\hat{x}_{t+1|t+1,\tilde{\Theta}}$ is determined similarly as in Eqn. (14), where the only difference is the next observation $y_{t+1,\Theta}$ replaced by $y^c_{t+1,\Theta}$.

## 5 Conclusion

In this paper, we make the first attempt to study the sim-to-real gap in continuous domain with partial observations. We show that the output policy of the robust adversarial algorithm enjoys the near-optimal sim-to-real gap, which depends on the intrinsic complexity of the simulator class. This work opens up several directions: Can we extend our results to the non-linear dynamics system? Can we design better sim-to-real algorithms beyond robust training? We leave them to future investigations.

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

# A  MISSING PARTS

## A.1  NOTATION TABLE

For the convenience of the reader, we summarize some notations that will be used.

| Notation | Explanation |
|---|---|
| $\mathscr{E}, \delta_{\mathscr{E}}$ | model class and its complexity |
| $\Theta = (A, B, C)$ | dynamics parameters defined in (1) |
| $V^{\star}(\Theta), J^{\star}(\Theta)$ | optimal cost of finite-horizon and infinite-horizon setting for $\Theta$ |
| $K, P, L, \Sigma$ | see (18), (17), (19), and (20) |
| $F$ | $AL$ (cf. (3)) |
| $\gamma_1, \kappa_2, \gamma_2$ | strongly stable coefficients |
| $\bar{X}_1, \bar{Y}, \bar{U}, \bar{X}_2$ | upper bounds of $\hat{x}_{t|t,\tilde{\Theta}_k}, y_t, u_t, \hat{x}_{t|t,\Theta}$ in (25), (26), and (27) |
| $N_S$ | $\max\{\|A\|, \|B\|, \|C\|\} \leq N_S$ for all $\Theta \in \mathscr{E}$ |
| $N_U$ | $\max\{\|Q\|_2, \|R\|_2\}$ |
| $N_K, N_P, N_\Sigma, N_L$ | upper bounds of $K, P, \Sigma, L$, respectively (see Appendix B) |
| $\mathcal{K}, \bar{\mathcal{K}}$ | number of episodes of LQG-VTR, upper bound of $\mathcal{K}$ (see Lemma 9) |
| $S(k), T(k)$ | start step and end step of episode $k$, respectively |
| $\tilde{\Theta}_t$ | optimistic model used for planning at step $t$ |
| $\tilde{\Theta}_k$ | optimistic model used during episode $k$ ($\tilde{\Theta}_t = \tilde{\Theta}_k$ for $S(k) \leq t \leq T(k)$) |
| $\hat{x}_{t|t,\Theta}$ | belief state at step $t$ measured under system $\Theta$ |
| $\bar{w}, \bar{z}$ | high probability upper bound of $\|w_t\|_2, \|z_t\|_2$ |
| $\bar{e}_t, \tilde{e}_t$ | Gaussian noises used in the induction stage 2 of the proof of Lemma 5 |
| $C_0, C_1, C_2$ | quantities defined in the induction proof of Lemma 5 |
| $T_A, T_B, T_C, T_L$ | lower bound of $T_w$ for desired properties at induction stage 1 |
| $T'_A, T'_B, T_{g0}, T_{g1}, T_{g2}, T_M$ | lower bound of $T_w$ for desired properties at induction stage 2 |
| $D_h$ | instance dependent constant used in reduction (see Lemma 3) |
| $a_0, b_0, c_0$ | instance dependent constant used to define $D_h$ |
| $D$ | the $\|\cdot\|_\infty$ norm of $\mathcal{F}$ (see Lemma 6) |
| $\Delta$ | the clipping error (difference between $f'$ and $f$, see Lemma 8) |

## A.2  ADDITIONAL DEFINITIONS

**Definition 2** (Controllablity & Observability). *A LQG system $\Theta = (A, B, C)$ is $(A, B)$ controllable if the controllability matrix $\begin{bmatrix} B & AB & A^2B & \ldots & A^{n-1}B \end{bmatrix}$ has full row rank. We say $(A, C)$ is observable if the observability matrix $\begin{bmatrix} C^\top & (CA)^\top & (CA^2)^\top & \ldots & (CA^{n-1})^\top \end{bmatrix}^\top$ has full column rank.*

**Definition 3** (Stability (Cohen et al., 2018; Lale et al., 2022)). *A LQG system $\Theta = (A, B, C)$ is stable if $\rho(A - BK(\Theta)) < 1$. It is strongly stable in terms of parameter $(\kappa, \gamma)$ if $\|K(\Theta)\|_2 \leq \kappa$ and there exists matrices $L$ and $H$ such that $A - BK = HLH^{-1}$ with $\|L\|_2 \leq 1 - \gamma, \|H\|_2\|H^{-1}\|_2 \leq \kappa$.*

*Note that any stable LQG system is also strongly stable in terms of some parameters (Cohen et al., 2018), and a strongly stable system is also stable.*

**Definition 4** (Covering Number). *We use $\mathcal{N}(\mathcal{F}, \epsilon)$ to denote the $\epsilon$-covering number of a set $\mathcal{F}$ with respect to the $\ell_\infty$ norm, which is the minimum integer $N$ such that their exists $\mathcal{F}' \in \mathcal{F}$ with $|\mathcal{F}'| = N$, and for any $f \in \mathcal{F}$ their exists $f' \in \mathcal{F}'$ satisfying $\|f - f'\|_\infty \leq \epsilon$.*

**Definition 5** ($\epsilon$-Independent). *For the function class $\mathcal{F}$ defined in $\mathcal{Z}$, we say $z$ is $\epsilon$-independent of $\{z_1, \cdots, z_n\} \in \mathcal{Z}$ if there exist $f, f' \in \mathcal{F}$ satisfying $\sqrt{\sum_{i=1}^n (f(z_i) - f'(z_i))^2} \leq \epsilon$ and $f(z) - f'(z) \geq \epsilon$.*

**Definition 6** (Eluder Dimension). *For the function class $\mathcal{F}$ defined in $\mathcal{Z}$, the $\epsilon$-Eluder dimension is the longest sequence $\{z_1, \cdots, z_n\} \in \mathcal{Z}$ such that there exists $\epsilon' \geq \epsilon$ where $z_i$ is $\epsilon'$-independent of $\{z_1, \cdots, z_{i-1}\}$ for any $i \in [n]$.*

### A.3 Preliminaries about the optimal control and the Kalman Filter

It is known that the optimal policy for this problem is a linear feedback control policy, i.e., $u_t = -K(\Theta)\hat{x}_{t|t,\Theta}$, where $K(\Theta)$ is the optimal control gain matrix and $\hat{x}_{t|t,\Theta}$ is the belief state at step $t$ (i.e. the estimation of the hidden state).

Let $P(\Theta)$ be the unique solution to the discrete-time algebraic Riccati equation (DARE):

$$P(\Theta) = A^\top P(\Theta)A + C^\top QC - A^\top P(\Theta)B(R + B^\top P(\Theta)B)^{-1}B^\top P(\Theta)A. \qquad (17)$$

Then $K(\Theta)$ can be calculated by

$$K(\Theta) = (R + B^\top P(\Theta)B)^{-1}B^\top P(\Theta)A. \qquad (18)$$

The belief state $\hat{x}_{t|t,\Theta}$ is defined as the mean of $x_t$ under system $\Theta$, which is decided by the system parameter $\Theta$ and history $\mathcal{H}_t$. Moreover, assuming $\hat{x}_{0|-1,\Theta} = 0$, the belief state can be calculated by the Kalman filter:

$$\hat{x}_{t|t,\Theta} = (I - L(\Theta)C)\hat{x}_{t|t-1,\Theta} + L(\Theta)y_t,$$
$$\hat{x}_{t|t-1,\Theta} = A\hat{x}_{t-1|t-1,\Theta} + Bu_{t-1}, \qquad (19)$$
$$L(\Theta) = \Sigma(\Theta)C^\top(C\Sigma(\Theta)C^\top + I)^{-1},$$

where $\Sigma(\Theta)$ is the **unique positive semidefinite** solution to the following DARE:

$$\Sigma(\Theta) = A\Sigma(\Theta)A^\top - A\Sigma(\Theta)C^\top(C\Sigma(\Theta)C^\top + I)^{-1}C\Sigma(\Theta)A^\top + I. \qquad (20)$$

We sometimes use $L, P, K, \Sigma$ as the shorthand of $L(\Theta), P(\Theta), K(\Theta), \Sigma(\Theta)$ when the system $\Theta$ is clear from the context.

The predictor form is another formulation of LQG instance (1) (Kalman, 1960; Lale et al., 2020b; 2021)

$$x_{t+1} = (A - F(\Theta)C)x_t + Bu_t + F(\Theta)y_t, \quad y_t = Cx_t + e_t, \qquad (21)$$

where $F(\Theta) = AL(\Theta)$ and $e_t$ denotes a zero-mean innovation process. In the steady state, we have $e_t \sim \mathcal{N}(\mathbf{0}, C\Sigma(\Theta)C^\top + I)$. The dynamics of $\hat{x}_{t|t-1,\Theta}$ follows exactly the predictor form.

### A.4 Domain Randomization

**Definition 7** (Domain Randomization Oracle (Chen et al., 2021))**.** *The domain randomization oracle returns a (history-dependent) policy $\pi_{\mathrm{DR}}$ such that*

$$\pi_{\mathrm{DR}} = \arg\min_{\pi} \mathbb{E}_{\Theta \sim d(\mathscr{E})}\left[V^\pi(\Theta) - V^\star(\Theta)\right], \qquad (22)$$

*where $V^\star$ is the optimal cost, $V^\pi$ is the cost of $\pi$, and $d(\mathscr{E})$ is a distribution over $\mathscr{E}$.*

As shown in Theorem 4 of Chen et al. (2021), even with an additional smooth assumption (Chen et al., 2021, Assumption 3), the performance of $\pi_{\mathrm{DR}}$ depends crucially on $d(\mathscr{E})$. In high dimensional continuous domain, the probability of sampling an accurate model close to the real world model by uniform randomization is exponentially small. Thus, the learner needs to carefully choose the domain randomization distribution $d(\mathscr{E})$ with strong prior knowledge of the real world model. In contrast, the robust adversarial training oracle does not worry about this, and it just needs to solve (7) by some min-max optimal control or robust RL algorithms.

## B   Uniform Bounded Simulator Set

Under Assumptions 1, 2, and 3, it is possible to provide a uniform upper bound on the spectral norm of $P(\Theta), \Sigma(\Theta), K(\Theta)$ and $L(\Theta)$ after the model selection procedure. Let $N_U \overset{\text{def}}{=} \max(\|Q\|_2, \|R\|_2)$.

For any $\Theta = (A, B, C) \in \mathscr{E}$, we have $\|K\|_2 \leq N_K$ by Assumption 2.

By definition $P$ is the unique solution to the equation

$$(A - BK)^\top P(A - BK) - P = C^\top QC + K^\top RK,$$

where $\rho(A - BK) < 1$. Therefore, Lemma B.4 of Simchowitz et al. (2020) shows

$$P = \sum_{k=0}^{\infty} \left((A - BK)^{\top}\right)^k (C^{\top} QC + K^{\top} RK)(A - BK)^k.$$

Since $\|(A - BK)^k\|_2 \le (1 - \gamma_1)^k$,

$$\|P\|_2 \le \left(N_S^2 N_U + N_K^2 N_U\right) \sum_{k=0}^{\infty} (1 - \gamma_1)^{2k} \le N_P \overset{\text{def}}{=} \frac{N_U(N_S^2 + N_K^2)}{2\gamma_1 - \gamma_1^2}.$$

Similarly, $\Sigma$ is the unique solution to the equation

$$(A - FC)\Sigma(A - FC)^{\top} - \Sigma = I + FF^{\top},$$

where $F = AL$ and $A - FC$ is $(\kappa_2, \gamma_2)$-strongly stable under Assumption 3. Thus we know

$$\|\Sigma\|_2 \le N_{\Sigma} \overset{\text{def}}{=} \frac{\kappa_2^2(1 + \kappa_2^2)}{2\gamma_2 - \gamma_2^2}.$$

Finally, we have

$$\|L\|_2 = \left\|\Sigma C^{\top}(C\Sigma C^{\top} + I)^{-1}\right\|_2 \le N_L \overset{\text{def}}{=} N_{\Sigma} N_S$$

since $\|(C\Sigma C^{\top} + I)^{-1}\|_2 \le 1$.

In this paper, we use $N_P, N_{\Sigma}, N_K, N_L$ as the uniform upper bound on the spectral norm of $P(\Theta), \Sigma(\Theta), K(\Theta)$ and $L(\Theta)$ for any $\Theta \in \mathscr{E}$.

## C  DETAILS FOR THE MODEL SELECTION PROCEDURE

In this section we provide a complete description of the model selection procedure.

The main purpose of running a model selection procedure at the beginning of LQG-VTR is to obtain a more accurate estimate $\hat{A}, \hat{B}, \hat{C}, \hat{L}$ of the real-world model $\Theta^{\star}$. This accurate estimate is very useful to show that the belief states encountered in LQG-VTR will be bounded (see Lemma 5).

We follow the warm up procedure used in Lale et al. (2021) to estimate Markov matrix $\mathbf{M}$, and perform the model selection afterwards. The matrix $\mathbf{M}$ is

$$\mathbf{M} = \begin{bmatrix} CF & C\bar{A}F & \ldots & C\bar{A}^{\tilde{H}-1}F & CB & C\bar{A}B & \ldots & C\bar{A}^{\tilde{H}-1}B \end{bmatrix},$$

where $\bar{A} = A - FC = A - ALC$.

---

**Algorithm 2** Model Selection

1: Input: model selection period length $T_w$ by Eqn. (56), simulator set $\mathscr{E}$.
2: Set $\sigma_u = 1$.
3: Execute action $u_t \sim \mathcal{N}(\mathbf{0}, \sigma_u^2 I)$ for $T_w$ steps, and gather dataset $\mathcal{D}_{\text{init}} = \{y_t, u_t\}_{t=1}^{T_w}$.
4: Set the truncation length $\tilde{H} = O(\log(\kappa_2 H) / \log(1/\gamma_2))$.
5: Estimate $\hat{\mathbf{M}}$ with $\mathcal{D}_{\text{init}}$ following Eqn. (11) of Lale et al. (2021).
6: Run SYSID$(\tilde{H}, \hat{G}, n)$ (Algorithm 2 of Lale et al. (2021)) to obtain an estimate $\hat{A}, \hat{B}, \hat{C}, \hat{L}$ of the real world system $\Theta^{\star}$.
7: Return $\mathscr{E} \cap \mathcal{C}$, where $\mathcal{C}$ is computed through Eqn. (23).

---

The regression model for $\mathbf{M}$ can be established by

$$y_t = \mathbf{M}\phi_t + e_t + C\bar{A}^{\tilde{H}} x_{t-\tilde{H}},$$

where $\phi_t = \begin{bmatrix} y_{t-1}^{\top} & \ldots & y_{t-H}^{\top} & u_{t-1}^{\top} & \ldots & u_{t-H}^{\top} \end{bmatrix}^{\top}$. Here $e_t$ is a Gaussian noise, and $C\bar{A}^H x_{t-H}$ is an exponentially small bias term.

Thanks to Assumption 3, $\bar{A}$ is a $(\kappa_2, \gamma_2)$-strongly stable matrix. Hence $\|\bar{A}^{\tilde{H}}\|_2 = O(\kappa_2(1-\gamma_2)^{\tilde{H}}) = O(1/H^2)$, and the bias term will be negligible.

Therefore, by estimating $\hat{M}$ with random action set $\mathcal{D}_{\text{init}}$, we come to a similar confidence set as shown in the Theorem 3.4 of Lale et al. (2021): There exists a unitary matrix $T \in \mathbb{R}^{n \times n}$ such that with probability at least $1 - \delta/4$, the real world model $\Theta^\star$ is contained in the set $\mathcal{C}$, where

$$
\mathcal{C} \stackrel{\text{def}}{=} \Big\{ \bar{\Theta} = (\bar{A}, \bar{B}, \bar{C}) :
$$
$$
\Big\| \hat{A} - T^\top \bar{A} T \Big\|_2 \le \beta_A, \Big\| \hat{B} - T^\top \bar{B} \Big\|_2 \le \beta_B, \Big\| \hat{C} - \bar{C} T \Big\|_2 \le \beta_C, \Big\| \hat{L} - T^\top L(\bar{\Theta}) \Big\|_2 \le \beta_L \Big\}.
\tag{23}
$$

Here the unitary $T$ is used because any LQG system transformed by a unitary matrix has exactly the same distribution of observation sequences as the original one (i.e., LQGs are equivalent under unitary transformations). Without loss of generality, we can assume $T = I$. $\beta_A, \beta_B, \beta_C, \beta_L$ measure the width of the confidence set, and we have

$$
\beta_A, \beta_B, \beta_C, \beta_L \le \sqrt{\frac{C_w}{T_w}}
\tag{24}
$$

for some instance-dependent parameter $C_w$ according to Theorem 3.4 of Lale et al. (2021).

Now we discuss how to decide the value of $T_w$ so that the following bounded state property holds after the model selection procedure, which is crucial to bound the regret of LQG-VTR. We define $\mathcal{K}$ as the number of episodes of LQG-VTR.

**Lemma 5** (Bounded State). *With probability at least $1 - \delta$, throughout the optimistic planning stage of LQG-VTR (i.e., after the model selection procedure), there exists constants $\bar{X}_1, \bar{X}_2, \bar{Y}, \bar{U}$ such that the belief state $\hat{x}_{t|t,\tilde{\Theta}_k}$ measured in $\tilde{\Theta}_k$ for any episode $k \in [\mathcal{K}]$ and any step $t$ in episode $k$ satisfy*

$$
\Big\| \hat{x}_{t|t,\tilde{\Theta}_k} \Big\|_2 \le \bar{X}_1.
\tag{25}
$$

*Accordingly for any step $t \in [H]$,*

$$
\|y_t\|_2 \le \bar{Y}, \quad \|u_t\|_2 \le \bar{U}.
\tag{26}
$$

*For any step $t \in [H]$ and any system $\Theta = (A, B, C) \in \mathcal{E}$, the belief state measured under $\Theta$ with history $(y_0, u_0, y_1, u_1, ..., y_{t-1}, u_{t-1}, y_t)$ is bounded*

$$
\big\| \hat{x}_{t|t,\Theta} \big\|_2 \le \bar{X}_2,
\tag{27}
$$

*where*

$$
\bar{X}_1, \bar{Y}, \bar{U}, \bar{X}_2 = O\left( (\sqrt{n} + \sqrt{p}) \log H \right).
\tag{28}
$$

*As a consequence, the algorithm does not terminate at Line 9 with probability $1 - \delta$.*

*Proof.* Let $S(k)$ and $T(k)$ be the starting step and ending step of episode $k$. For simplicity, we use $\tilde{\Theta}_t = (\tilde{A}_t, \tilde{B}_t, \tilde{C}_t)$ to denote $\tilde{\Theta}_k$ for $S(k) \le t \le T(k)$ for time step $t$ (The first episode starts after the model selection procedure, $S(1) = T_w + 1$). The core problem here is to show that after the model selection procedure (Algorithm 2), the belief state $\hat{x}_{t|t,\tilde{\Theta}_t}$ under $\tilde{\Theta}_t$ stays bounded, then $u_t$ also stays bounded since $u_t = -K(\tilde{\Theta}_t)\hat{x}_{t|t,\tilde{\Theta}_t}$. Since $\Theta^\star \in \mathcal{C}$ with probability at least $1 - \delta/4$, we assume this event happens.

Before going through the proof, we define constant $\bar{w}, \bar{z}$ and $\bar{u}$ such that with probability $1 - \delta/4$, $\|w_t\|_2 \le \bar{w}, \|z_t\|_2 \le \bar{z}$ for any $0 \le t \le H$, and $\|u_t\|_2 \le \bar{u}$ for $0 \le t \le T_w$. Since $w_t, z_t, u_t$ are independent Gaussian variables, we know $\bar{w} = O(\sqrt{n \log(H)}), \bar{z} = O(\sqrt{p \log(H)}), \bar{u} = O(\sqrt{m \log(T_w)})$. For the model selection procedure (which contains the first $T_w$ steps), it holds that with probability $1 - \delta/4$, for any $t \le T_w$, $\|x_t\|_2 \le O(\sqrt{n \log(H)}), \|u_t\|_2 \le \bar{u}, \|y_t\|_2 \le O(N_S \sqrt{n \log(H)} + \sqrt{p \log(H)})$ by Lemma D.1 of Lale et al. (2020c). We assume this high probability event happens for now.

We use an inductive argument to show that for any episode $k$ and step $t$ therein, it holds that $\|\hat{x}_{t|t,\tilde{\Theta}_t}\|_2 \le C_0 (1 + 1/\bar{\mathcal{K}})^k$ for some $C_0$. To guarantee action $u_t$ does not blow up, we set the

upper bound of $\|\hat{x}_{t|t,\tilde{\Theta}_t}\|_2$ as $\bar{X}_1 \stackrel{\text{def}}{=} eC_0 \geq C_0(1 + 1/\bar{\mathcal{K}})^k$ for any $k \leq \mathcal{K} < \bar{\mathcal{K}}$. Thus, LQG-VTR terminates as long as $\|\hat{x}_{t|t,\tilde{\Theta}_t}\|_2$ exceeds that upper bound (Line 9 of Algorithm 1). For convenient, we call the model selection procedure as *episode 0* (so $T(0) = T_w$). As the inductive hypothesis, suppose that $\|\hat{x}_{t|t,\tilde{\Theta}_t}\|_2 \leq D_0 \leq C_0(1 + 1/\bar{\mathcal{K}})^k$ holds for some $D_0$ and step $t \leq T(k) + 1$ when $k \geq 1$. We define $D_0 = C_0$ for $C_0 = O((\sqrt{n} + \sqrt{p}) \log H)$ specified later when $k = 0$. Then we know that the algorithm does not terminate in the first $k$ episodes. Moreover, we have for any $t \leq T(k)$ it holds that $\|u_t\|_2 \leq N_K D_0$. Thus for $t \leq T(k) + 1$, by definition of $x_t$ and $y_t$ we have

$$\|x_t\|_2 \leq \frac{\Phi(A^\star)}{1 - \rho(A^\star)} \cdot (N_S N_K D_0 + \bar{w}), \|y_t\|_2 \leq N_S \|\bar{x}_t\|_2 + \bar{z}. \tag{29}$$

Suppose $D_0 \geq \bar{w}\Phi(A^\star)/(1 - \rho(A^\star))$, so that

$$\|x_t\|_2 \leq \left(1 + \frac{N_S N_K \Phi(A^\star)}{1 - \rho(A^\star)}\right) D_0, \|y_t\|_2 \leq \left(1 + N_S + \frac{N_S^2 N_K \Phi(A^\star)}{1 - \rho(A^\star)}\right) D_0. \tag{30}$$

Hence, we can define three constants $D_u, D_x, D_y$ such that for any $t \leq T(k)$ it holds that $\|u_t\|_2 \leq D_u D_0$; for any $t \leq T(k) + 1$, we have $\|x_t\|_2 \leq D_x D_0, \|y_t\|_2 \leq D_y D_0$. Note that $D_u, D_x, D_y$ are all instance-dependent constants. To be more concrete, it suffices to set

$$D_u = N_K \quad D_x = 1 + \frac{N_S N_K \Phi(A^\star)}{1 - \rho(A^\star)} \quad D_y = 1 + N_S + \frac{N_S^2 N_K \Phi(A^\star)}{1 - \rho(A^\star)}.$$

**INDUCTION STAGE 1**: FIX $t = T(k) + 1$ AND SHOW $\|\hat{x}_{t|t,\tilde{\Theta}_k}\|_2$ IS VERY CLOSE TO $\|\hat{x}_{t|t,\tilde{\Theta}_{k+1}}\|_2$

In stage 1, we need to derive four terms $T_A, T_B, T_C, T_L$ as the lower bound of $T_w$ to achieve the desired concentration property. This means as long as $T_w \geq T_A, T_B, T_C, T_L$, we can prove the induction stage 1 thanks to the tight confidence set $\mathcal{C}$ defined in Eqn. (23). Now fix $t = T(k) + 1 = S(k+1)$, for any $\Theta \in \mathcal{U}_1$ we have

$$\hat{x}_{t|t,\Theta} = \sum_{s=1}^{t} \mathcal{A}^{s-1}\left(\mathcal{B}u_{t-s} + Ly_{t-s+1}\right) + \mathcal{A}^t Ly_0, \tag{31}$$

where $\mathcal{A} \stackrel{\text{def}}{=} (I - LC)A, \mathcal{B} \stackrel{\text{def}}{=} (I - LC)B$. Note that $\|\mathcal{A}^s\|_2 \leq \kappa_2(1 - \gamma_2)^s$ according to Assumption 3. Therefore, for $\Theta = \tilde{\Theta}_k$ and $\Theta = \tilde{\Theta}_{k+1}$ (both $\tilde{\Theta}_k$ and $\tilde{\Theta}_{k+1}$ is chosen from the confidence set $\mathcal{C}$ by definition),

$$\hat{x}_{t|t,\tilde{\Theta}_k} - \hat{x}_{t|t,\tilde{\Theta}_{k+1}}$$
$$= \sum_{s=1}^{t} \tilde{\mathcal{A}}_k^{s-1}\left(\tilde{\mathcal{B}}_k u_{t-s} + \tilde{L}_k y_{t-s+1}\right) + \tilde{\mathcal{A}}_k^t \tilde{L}_k y_0 - \sum_{s=1}^{t} \tilde{\mathcal{A}}_{k+1}^{s-1}\left(\tilde{\mathcal{B}}_{k+1} u_{t-s} + \tilde{L}_{k+1} y_{t-s+1}\right) + \tilde{\mathcal{A}}_{k+1}^t \tilde{L}_{k+1} y_0.$$

To move on,

$$\hat{x}_{t|t,\tilde{\Theta}_k} - \hat{x}_{t|t,\tilde{\Theta}_{k+1}}$$
$$= \sum_{s=1}^{t} \tilde{\mathcal{A}}_k^{s-1}\left(\tilde{\mathcal{B}}_k u_{t-s} + \tilde{L}_k y_{t-s+1}\right) - \sum_{s=1}^{t} \tilde{\mathcal{A}}_k^{s-1}\left(\tilde{\mathcal{B}}_{k+1} u_{t-s} + \tilde{L}_{k+1} y_{t-s+1}\right)$$
$$+ \sum_{s=1}^{t} \tilde{\mathcal{A}}_k^{s-1}\left(\tilde{\mathcal{B}}_{k+1} u_{t-s} + \tilde{L}_{k+1} y_{t-s+1}\right) - \sum_{s=1}^{t} \tilde{\mathcal{A}}_{k+1}^{s-1}\left(\tilde{\mathcal{B}}_{k+1} u_{t-s} + \tilde{L}_{k+1} y_{t-s+1}\right)$$
$$+ \tilde{\mathcal{A}}_k^t \tilde{L}_k y_0 - \tilde{\mathcal{A}}_{k+1}^t \tilde{L}_{k+1} y_0.$$

We bound these three terms separately in the following. For the first term, observe that when $T_w \geq T_L$, we have

$$\left\|\sum_{s=1}^{t} \tilde{\mathcal{A}}_k^{s-1}(\tilde{L}_k - \tilde{L}_{k+1})y_{t-s+1}\right\|_2 \leq \sum_{s=1}^{t} \kappa_2(1 - \gamma_2)^{s-1} \cdot \frac{2\sqrt{C_w}}{\sqrt{T_L}} \cdot D_y D_0$$
$$\leq \frac{2\kappa_2 D_y \sqrt{C_w}}{\gamma_2 \sqrt{T_L}} \cdot D_0. \tag{32}$$

where the first inequality is due to the confidence set (Eqn. (23)), the induction hypothesis, and Assumption 3. Thus, as long as $T_L \geq (12\kappa_2 D_y \sqrt{C_w}\bar{\mathcal{K}}/\gamma_2)^2 = O(\bar{\mathcal{K}}^2)$, we know

$$\left\| \sum_{s=1}^{t} \tilde{\mathcal{A}}_k^{s-1}(\tilde{L}_k - \tilde{L}_{k+1})y_{t-s+1} \right\|_2 \leq \frac{D_0}{6\bar{\mathcal{K}}}. \tag{33}$$

Similarly, we have

$$\left\| \tilde{\mathcal{B}}_k - \tilde{\mathcal{B}}_{k+1} \right\|_2 \leq \sqrt{C_w} \left( \frac{1 + N_L N_S}{\sqrt{T_B}} + \frac{N_S^2}{\sqrt{T_L}} + \frac{N_L N_S}{\sqrt{T_C}} \right)$$

by checking the definition of $\tilde{\mathcal{B}}_k$ and $\tilde{\mathcal{B}}_{k+1}$.

Therefore, as long as

$$T_B \geq \left( \frac{36\kappa_2 D_u \sqrt{C_w}(1 + N_L N_S)\bar{\mathcal{K}}}{\gamma_2} \right)^2 = O(\bar{\mathcal{K}}^2)$$

$$T_L \geq \left( \frac{36\kappa_2 D_u \sqrt{C_w}N_S^2\bar{\mathcal{K}}}{\gamma_2} \right)^2 = O(\bar{\mathcal{K}}^2)$$

$$T_C \geq \left( \frac{36\kappa_2 D_u \sqrt{C_w}N_L N_S\bar{\mathcal{K}}}{\gamma_2} \right)^2 = O(\bar{\mathcal{K}}^2),$$

we have

$$\left\| \tilde{\mathcal{B}}_k - \tilde{\mathcal{B}}_{k+1} \right\|_2 \leq \frac{\gamma_2}{12\kappa_2 D_u\bar{\mathcal{K}}}.$$

Therefore, it holds that

$$\left\| \sum_{s=1}^{t} \tilde{\mathcal{A}}_k^{s-1}(\tilde{\mathcal{B}}_k - \tilde{\mathcal{B}}_{k+1})u_{t-s} \right\|_2 \leq \frac{D_0}{6\bar{\mathcal{K}}}. \tag{34}$$

following the same reason as Eqn. (32).

To sum up, we require $T_B \geq O(\bar{\mathcal{K}}^2), T_L \geq O(\bar{\mathcal{K}}^2), T_C \geq O(\bar{\mathcal{K}}^2)$ to ensure that as long as $T_w \geq \max(T_B, T_L, T_C)$, the first term is bounded by

$$\left\| \sum_{s=1}^{t} \tilde{\mathcal{A}}_k^{s-1} \left( (\tilde{\mathcal{B}}_k - \tilde{\mathcal{B}}_{k+1})u_{t-s} + (\tilde{L}_k - \tilde{L}_{k+1})y_{t-s+1} \right) \right\|_2 \leq \frac{D_0}{3\bar{\mathcal{K}}} \tag{35}$$

from Eqn. (33) and Eqn. (34).

For the second term,

$$\left\| \sum_{s=1}^{t} \left( \tilde{\mathcal{A}}_k^{s-1} - \tilde{\mathcal{A}}_{k+1}^{s-1} \right) \left( \tilde{\mathcal{B}}_{k+1}u_{t-s} + \tilde{L}_{k+1}y_{t-s+1} \right) \right\| \tag{36}$$

$$\leq \sum_{s=1}^{t} \xi_k \cdot (s-1)\kappa_2^2(1-\gamma_2)^{s-2} \left( (N_S + N_S^2 N_L)D_u + N_L D_y \right) D_0 \quad \text{(see Eqn. (148))} \tag{37}$$

$$\leq \xi_k \cdot (\kappa_2/\gamma_2)^2 t(1-\gamma_2)^{t-1} \left( (N_S + N_S^2 N_L)D_u + N_L D_y \right) D_0, \tag{38}$$

where

$$\xi_k = \left\| \tilde{\mathcal{A}}_k - \tilde{\mathcal{A}}_{k+1} \right\|_2.$$

Note that $C_1 \stackrel{\text{def}}{=} \max_{t \geq 0} t(1-\gamma_2)^{t-1}$ is a constant. Similarly, we require $T_A, T_C, T_L \geq O(\bar{\mathcal{K}}^2)$ (see Eqn. (32) and Eqn. (33) for an example on how $T_A, T_C, T_L$ are derived) to ensure that as long as $T_w \geq \max(T_A, T_C, T_L)$, it holds that

$$\xi_k \cdot (\kappa_2/\gamma_2)^2 C_1 \left( (N_S + N_S^2 N_L)D_u + N_L D_y \right) D_0 \leq \frac{D_0}{3\bar{\mathcal{K}}}. \tag{39}$$

Using the same argument, by setting an appropriate $T_L$ we have as long as $T_w \geq T_L$, it holds that

$$\left\| \tilde{\mathcal{A}}_k^t \tilde{L}_k y_0 - \tilde{\mathcal{A}}_{k+1}^t \tilde{L}_{k+1} y_0 \right\| \leq \frac{D_0}{3\overline{\mathcal{K}}}. \tag{40}$$

As a consequence, combining Eqn. (35), (39), and (40) gives

$$\left\| \hat{x}_{t|t,\tilde{\Theta}_k} - \hat{x}_{t|t,\tilde{\Theta}_{k+1}} \right\|_2 \leq \frac{D_0}{\overline{\mathcal{K}}} \tag{41}$$

$$\left\| \hat{x}_{t|t,\tilde{\Theta}_{k+1}} \right\|_2 \leq D_0 \left( 1 + \frac{1}{\overline{\mathcal{K}}} \right). \tag{42}$$

For now we have proved the induction at the beginning step $t = S(k+1)$ of episode $k+1$, it suffices to show that $\|\hat{x}_{t|t,\tilde{\Theta}_{k+1}}\|_2 \leq (1 + 1/\overline{\mathcal{K}})D_0$ for the rest of episode $k + 1$.

**INDUCTION STAGE 2**: SHOW THAT $\|\hat{x}_{t|t,\tilde{\Theta}_{k+1}}\|_2 \leq (1 + 1/\overline{\mathcal{K}})D_0$ HOLDS FOR THE WHOLE EPISODE $k+1$.

Suppose LQG-VTR does not terminate during episode $k+1$, then the algorithms follows the optimal control of $\tilde{\Theta}_{k+1}$. Denote $\tilde{\Theta}_{k+1} = (\tilde{A}, \tilde{B}, \tilde{C})$, we follow the Eqn. (46) of Lale et al. (2020c) to decompose $\hat{x}_{t|t,\tilde{\Theta}_{k+1}}$ for $S(k+1) + 1 \leq t \leq T(k+1) + 1$:

$$\hat{x}_{t|t,\tilde{\Theta}_{k+1}} = M\hat{x}_{t-1|t-1,\tilde{\Theta}_{k+1}} + \tilde{L}C^\star \left( \hat{x}_{t|t-1,\Theta^\star} - \hat{x}_{t|t-1,\tilde{\Theta}} \right) + \tilde{L}z_t + \tilde{L}C^\star \left( x_t - \hat{x}_{t|t-1,\Theta^\star} \right), \tag{43}$$

where $M \stackrel{\text{def}}{=} (\tilde{A} - \tilde{B}\tilde{K} - \tilde{L}(\tilde{C}\tilde{A} - \tilde{C}\tilde{B}\tilde{K} - C^\star\tilde{A} + C^\star\tilde{B}\tilde{K}))$. The main idea to prove induction stage 2 is to show that $\hat{x}_{t|t,\tilde{\Theta}_{k+1}}$ will also be bounded by $(1 + 1/\overline{\mathcal{K}})D_0$ as long as $\hat{x}_{t-1|t-1,\tilde{\Theta}_{k+1}}$ is bounded by $(1 + 1/\overline{\mathcal{K}})D_0$, given the conclusion $\hat{x}_{S(k+1)|S(k+1),\tilde{\Theta}_{k+1}} \leq (1 + 1/\overline{\mathcal{K}})D_0$ of the induction stage 1. To achieve this end, we divide the induction stage 2 into 2 phases. We first show that $\hat{x}_{t|t-1,\Theta^\star} - \hat{x}_{t|t-1,\tilde{\Theta}}$ is small in the first phase, and prove the above main idea formally in the second phase.

Before coming to the main proof for induction stage 2, we first observe that $\bar{e}_t \stackrel{\text{def}}{=} z_t + C^\star \left( x_t - \hat{x}_{t|t-1,\Theta^\star} \right)$ is a $(N_S N_\Sigma + 1)$-subGaussian noise, so with probability at least $1 - \delta/4$ we know $\|\bar{e}_t\|_2$ is bounded by $C_2(N_S N_\Sigma + 1)\sqrt{n \log(H)}$ for any $1 \leq t \leq H$ and some constant $C_2$. Assume this high probability event happens for now.

**Phase 1: bound** $\hat{x}_{t|t-1,\Theta^\star} - \hat{x}_{t|t-1,\tilde{\Theta}}$  We mainly follow the decomposition and induction techniques in Lale et al. (2020c) to finish phase 1. However, we note that our analysis here is much more complicated than the analysis in Lale et al. (2020c), because they have *only one* commit episode after the pure exploration stage (the agent chooses random actions in their pure exploration stage, just the same as the model selection procedure in LQG-VTR) but we have multiple episodes here.

Define $\Delta_t \stackrel{\text{def}}{=} \hat{x}_{t+S(k+1)|t+S(k+1)-1,\Theta^\star} - \hat{x}_{t+S(k+1)|t+S(k+1)-1,\tilde{\Theta}_{k+1}}$ for $1 \leq t \leq T(k+1) - S(k+1) + 1$. For $t = 1$, we have

$$\begin{aligned}
\Delta_1 &= \hat{x}_{S(k+1)+1|S(k+1),\Theta^\star} - \hat{x}_{S(k+1)+1|S(k+1),\tilde{\Theta}_{k+1}} \\
&= A^\star \hat{x}_{S(k+1)|S(k+1),\Theta^\star} + B^\star u_{S(k+1)} - \tilde{A}\hat{x}_{S(k+1)|S(k+1),\tilde{\Theta}_{k+1}} - \tilde{B}u_{S(k+1)}.
\end{aligned} \tag{44}$$

Since we have assumed $\Theta^\star \in \mathcal{C}$, we know $\|\hat{x}_{S(k+1)|S(k+1),\Theta^\star} - \hat{x}_{S(k+1)|S(k+1),\tilde{\Theta}_{k+1}}\|_2$ will be small according to induction stage 1. Specifically, we can set $T'_A, T'_B \geq O((N_L N_S \kappa_3)^2/(1 - \gamma_1)^2)$ so that as long as $T_w \geq T'_A, T'_B$, we have $\|\Delta_1\|_2 \leq D_0(1 - \gamma_1)/(8N_L N_S \kappa_3)$ for a constant $\kappa_3$ defined later. Now that we have come up with an upper bound on $\|\Delta_1\|_2$, we hope to bound each $\|\Delta_t\|_2$ and prove Eqn. (47).

For $1 < t \leq T(k+1) - S(k+1) + 1$, we follow Eqn. (49) and Eqn. (50) of Lale et al. (2020c) to perform the decomposition

$$\Delta_t = G_0^{t-1}\Delta_1 + G_1 \sum_{j=1}^{t-1} G_0^{t-1-j} \left( G_2^{j-1}\hat{x}_{S(k+1)+1|S(k+1),\Theta^\star} + \sum_{s=1}^{j-1} G_2^{j-s-1}G_3\Delta_s \right) + \tilde{e}_t, \tag{45}$$

where $\tilde{e}_t$ is a noise term, whose $l_2$-norm is bounded by $O(\sqrt{p\log(H)})$ with probability $1 - \delta/4$ for any episode $k$ and step $t$. We again assume this high probability event happens for now. We define $\Lambda_{\tilde{\Theta}} \stackrel{\text{def}}{=} \tilde{A} - A^\star - \tilde{B}\tilde{K} + B^\star\tilde{K}$ and thus

$$
\begin{aligned}
G_0 &= (A^\star + \Lambda_{\tilde{\Theta}})(I - \tilde{L}\tilde{C}), \\
G_1 &= (A^\star + \Lambda_{\tilde{\Theta}})\tilde{L}(\tilde{C} - C^\star) - \Lambda_{\tilde{\Theta}}, \\
G_2 &= A^\star - B^\star\tilde{K} + B^\star\tilde{K}\tilde{L}(\tilde{C} - C^\star), \\
G_3 &= (A^\star - B^\star\tilde{K})L^\star + B^\star\tilde{K}(L^\star - \tilde{L}).
\end{aligned}
\tag{46}
$$

Using a similar argument as in Lale et al. (2020c), we define $T_{g0}, T_{g2}$ so that $G_0, G_2$ are $(\kappa_3, \gamma_3)$-strongly stable for some $\gamma_3 \le \min(\gamma_2/2, (1 - \gamma_1)/2)$ and $\kappa_3 \ge 1$ as long as $T_w \ge \max(T_{g0}, T_{g2})$. This is possible because as long as $\tilde{\Theta}$ is close enough to $\Theta$, $G_0$ will be strongly stable according to Assumption 3 while $G_2$ will be contractible in that $\|G_2\|_2 \le (1 + \gamma_1)/2$ according to Assumption 2. To be more concrete, we show how to construct $T_{g2}$ in the following, and the construction of $T_{g0}$ are analogous to that of $T_{g2}$.

Observe that

$$
G_2 - \left(\tilde{A} - \tilde{B}\tilde{K}\right) = A^\star - \tilde{A} + (\tilde{B} - B^\star)\tilde{K} + B^\star\tilde{K}\tilde{L}(\tilde{C} - C^\star).
$$

By a similar argument used in Eqn. (32) and Eqn. (33), we know

$$
\left\| A^\star - \tilde{A} + (\tilde{B} - B^\star)\tilde{K} + B^\star\tilde{K}\tilde{L}(\tilde{C} - C^\star) \right\|_2 \le \frac{1 - \gamma_1}{2},
$$

as long as $T_w \ge T_{g2} \stackrel{\text{def}}{=} 36C_w N_S^2 N_K^2 N_L^2/(1 - \gamma_1)^2$. This further implies

$$
\|G_2\|_2 \le \|\tilde{A} - \tilde{B}\tilde{K}\|_2 + \frac{1 - \gamma_1}{2} \le \frac{1 + \gamma_1}{2},
$$

where the second inequality is by Assumption 3.

Now we prove that

$$
\|\Delta_t\|_2 \le \frac{1 - \gamma_1}{4N_L N_S} D_0
\tag{47}
$$

holds for all $1 \le t \le T(k + 1) - S(k + 1) + 1$.

First of all, we know $\|\Delta_1\|_2 \le D_0(1 - \gamma_1)/(8N_L N_S \kappa_3) \le (1 - \gamma_1)D_0/(4N_L N_S)$ according to Eqn. (44) and $\kappa_3 \ge 1$. For any fixed $t$, suppose $\|\Delta_s\|_2 \le (1 - \gamma_1)D_0/(4N_L N_S)$ for all $1 \le s \le t - 1$, then the decomposition equation (45) implies

$$
\begin{aligned}
\|\Delta_t\|_2 &= \left\| G_0^{t-1}\Delta_1 + G_1 \sum_{j=1}^{t-1} G_0^{t-1-j}\left( G_2^{j-1}\hat{x}_{S(k+1)+1|S(k+1),\Theta^\star} + \sum_{s=1}^{j-1} G_2^{j-s-1}G_3\Delta_s \right) + \tilde{e}_t \right\|_2 \\
&\le \left\| G_1 \sum_{j=1}^{t-1} G_0^{t-1-j}\left( G_2^{j-1}\hat{x}_{S(k+1)+1|S(k+1),\Theta^\star} + \sum_{s=1}^{j-1} G_2^{j-s-1}G_3\Delta_s \right) \right\|_2 \\
&\quad + \kappa_3(1 - \gamma_3)^{t-1}\|\Delta_1\|_2 + \|\tilde{e}_t\|_2 \\
&\le \|G_1\|_2 \cdot (t - 1)\kappa_3^2(1 - \gamma_3)^{t-1} \cdot 2N_S(1 + N_K)D_0 + \left\| G_1 \sum_{j=1}^{t-1} G_0^{t-1-j} \sum_{s=1}^{j-1} G_2^{j-s-1}G_3\Delta_s \right\|_2 \\
&\quad + \kappa_3(1 - \gamma_3)^{t-1}\|\Delta_1\|_2 + \|\tilde{e}_t\|_2 \\
&\le \|G_1\|_2 \cdot (t - 1)\kappa_3^2(1 - \gamma_3)^{t-1} \cdot 2N_S(1 + N_K)D_0 + \left(\frac{\kappa_3}{\gamma_3}\right)^2 \|G_1 G_3\|_2 \cdot \frac{1 - \gamma_1}{4N_L N_S} D_0 \\
&\quad + \kappa_3(1 - \gamma_3)^{t-1}\|\Delta_1\|_2 + \|\tilde{e}_t\|_2.
\end{aligned}
$$

The first inequality follows from the strong stability of $G_0$ and the concentration of noises $\tilde{e}_t$. The second inequality is due to the strong stability of $G_0, G_2$, and $\|\hat{x}_{S(k+1)+1|S(k+1),\Theta^\star}\|_2 \le (1 +$

$N_K)(N_S + N_S/\bar{\mathcal{K}})D_0 \leq 2N_S(1 + N_K)D_0$ according to induction stage 1. The last inequality is because the assumption that $\|\Delta_s\|_2 \leq (1 - \gamma_1)D_0/(4N_L N_S)$ for all $1 \leq s \leq t - 1$.

When $T_w \geq T_{g1}$ for some $T_{g1} \geq 1$ we know that

$$\|G_1\|_2 \leq \left(N_S + \frac{\sqrt{C_w}(1 + N_K)}{\sqrt{T_{g1}}}\right) \frac{N_L \sqrt{C_w}}{\sqrt{T_{g1}}} + \frac{\sqrt{C_w}(1 + N_K)}{\sqrt{T_{g1}}}$$

$$\leq \frac{\sqrt{C_w}\left(N_L N_S + (\sqrt{C_w}N_L + 1)(1 + N_K)\right)}{\sqrt{T_{g1}}}.$$

Since $\max_{t \geq 0}(t - 1)(1 - \gamma_3)^{t-1}$ and $\|G_3\|_2$ are both (instance-dependent) constants, there exists an instance-dependent constant $T_{g1}$ that for $T_w \geq T_{g1}$, it holds that

$$\|G_1\|_2 \cdot (t - 1)\kappa_3^2(1 - \gamma_3)^{t-1} \cdot 2N_S(1 + N_K)D_0 \leq \frac{1 - \gamma_1}{24N_L N_S}D_0. \tag{48}$$

$$\|G_1\|_2 \cdot \left(\frac{\kappa_3}{\gamma_3}\right)^2 \|G_3\|_2 \cdot \frac{1 - \gamma_1}{4N_L N_S}D_0 \leq \frac{1 - \gamma_1}{24N_L N_S}D_0. \tag{49}$$

Moreover, the bound on $\|\Delta_1\|_2$ from Eqn. (44) implies that

$$\kappa_3(1 - \gamma_3)^{t-1}\|\Delta_1\|_2 \leq \frac{1 - \gamma_1}{8N_L N_S}D_0. \tag{50}$$

Note that we can set $C_0$ so that $D_0$ is large enough to guarantee

$$\frac{(1 - \gamma_1)D_0}{24N_L N_S} \geq \|\tilde{e}_t\|_2 = O(\sqrt{p\log(H)}) \tag{51}$$

for any episode $k$ and step $t$ in it.

The proof of Eqn. (47) can be obtained by combining Eqn. (48), (49), (50), and (51).

As a final remark, $T_{g0}, T_{g1}, T_{g2}$ has no dependency on $H$.

**Phase 2: finish induction stage 2**  Now we come back to the main decomposition formula (43). Define $T_M = O(2^2/(1 - \gamma_1)^2)$ so that $\|C - \tilde{C}\|_2$ is small enough to guarantee $\|M\|_2 \leq (1 + \gamma_1)/2$ (see the definition of $M$ at the beginning of induction stage 2) as long as $T_w \geq T_M$. Suppose we know $\|\hat{x}_{t-1|t-1, \tilde{\Theta}_{k+1}}\|_2 \leq (1 + 1/\bar{\mathcal{K}})D_0$ for time step $t - 1$, then as long as $D_0 \geq 4C_2 N_L(N_S N_\Sigma + 1)\sqrt{n\log(H)}/(1 - \gamma_1)$, we have

$$\left\|\hat{x}_{t|t, \tilde{\Theta}_{k+1}}\right\|_2 \leq \|M\|_2 \left(1 + \frac{1}{\bar{\mathcal{K}}}\right)D_0 + \tilde{L}C^\star\left\|\Delta_{t-S(k+1)}\right\|_2 + \tilde{L}\bar{e}_t \tag{52}$$

$$\leq \frac{1 + \gamma_1}{2}\left(1 + \frac{1}{\bar{\mathcal{K}}}\right)D_0 + \frac{1 - \gamma_1}{4}D_0 + \frac{1 - \gamma_1}{4}D_0 \tag{53}$$

$$\leq \left(1 + \frac{1}{\bar{\mathcal{K}}}\right)D_0. \tag{54}$$

Fortunately, we know that $\|\hat{x}_{S(k+1)|S(k+1), \tilde{\Theta}_{k+1}}\|_2 \leq (1 + 1/\bar{\mathcal{K}})D_0$ according to induction stage 1. Using the recursion argument above, we can show that $\|\hat{x}_{t|t, \tilde{\Theta}_{k+1}}\|_2 \leq (1 + 1/\bar{\mathcal{K}})D_0$ holds for $S(k + 1) \leq t \leq T(k + 1) + 1$.

We have proved the induction from episode $k$ to $k + 1$ so far under the condition that LQG-VTR does not terminate at Line 9 during episode $k$. On the other hand, if for some step $S(k + 1) + 1 \leq t_0 \leq T(k + 1)$ the algorithm entered Line 9 (note that it cannot be $t_0 = S(k + 1)$ because induction stage 1 does not depend on whether the algorithm terminates), then the decomposition for $\hat{x}_{t|t, \tilde{\Theta}_{k+1}}$ (43) and decomposition for $\Delta_t$ (45) still works for steps $t < t_0$. Therefore, there must be some high probability event fails at step $t_0 - 1$ (i.e., $\bar{e}_{t_0-1}$ and/or $\tilde{e}_{t_0-1}$ explode). This is impossible since we have assumed that all the high probability events happen.

Thus under the condition that all the high probability events happen, it suffices to choose the constant $C_0 = O((\sqrt{n} + \sqrt{p}) \log H)$ so that Eqn. (30), (51), (52) all hold, and LQG-VTR does not terminate at Line 9 due to the definition of $\bar{X}_1 = eC_0$. Since these high probability events happen with probability $1 - \delta$ and recall that the total number of episodes is bounded by $\bar{\mathcal{K}}$, we know that with probability at least $1 - \delta$,

$$\|\hat{x}_{t|t,\tilde{\Theta}_t}\|_2 \le C_0 \left(1 + \frac{1}{\bar{\mathcal{K}}}\right)^{\bar{\mathcal{K}}} \le eC_0 = \bar{X}_1 = O((\sqrt{n} + \sqrt{p}) \log H). \tag{55}$$

As a result, we know $\|u_t\|_2 \le \bar{U} \stackrel{\text{def}}{=} N_K \bar{X}_1$ because $u_t = -K(\tilde{\Theta}_t)\hat{x}_{t|t,\tilde{\Theta}_t}$. Moreover,

$$
\begin{aligned}
y_t &= C^\star x_t + z_t = C^\star \hat{x}_{t|t-1,\tilde{\Theta}_{t-1}} + C^\star \left(x_t - \hat{x}_{t|t-1,\tilde{\Theta}_{t-1}}\right) + z_t \\
&= C^\star \left(\tilde{A}_{t-1}\hat{x}_{t-1|t-1,\tilde{\Theta}_{t-1}} + \tilde{B}_{t-1}u_{t-1}\right) + C^\star \left(x_t - \hat{x}_{t|t-1,\tilde{\Theta}_{t-1}}\right) + z_t \\
&= C^\star \left(\tilde{A}_{t-1}\hat{x}_{t-1|t-1,\tilde{\Theta}_{t-1}} + \tilde{B}_{t-1}u_{t-1}\right) \\
&\quad + C^\star \left(x_t - \hat{x}_{t|t-1,\Theta^\star} + \hat{x}_{t|t-1,\Theta^\star} - \hat{x}_{t|t-1,\tilde{\Theta}_{t-1}}\right) + z_t.
\end{aligned}
$$

Observe that $\|u_{t-1}\|_2 \le \bar{U}$, the $l_2$-norm of noise term $x_t - \hat{x}_{t|t-1,\Theta^\star}$ is bounded by $C_2(N_S N_\Sigma + 1)\sqrt{n \log(H)}$, and the last term $\hat{x}_{t|t-1,\Theta^\star} - \hat{x}_{t|t-1,\tilde{\Theta}_{t-1}}$ can be bounded through Eqn. (47), we have

$$\|y_t\|_2 \le \bar{Y} \stackrel{\text{def}}{=} N_S^2(\bar{X}_1 + \bar{U}) + C_2(N_S N_\Sigma + 1)\sqrt{n \log(H)} + \frac{(1 - \gamma_1)\bar{X}_1}{4N_L}.$$

Note that $\hat{x}_{0|0,\Theta} = L(\Theta)y_0$, by Eqn. (31)

$$
\begin{aligned}
\left\|\hat{x}_{t|t,\Theta}\right\|_2 &\le \kappa_2(1 - \gamma_2)^t N_L \bar{Y} + \sum_{s=1}^{t} \kappa_2(1 - \gamma_2)^{s-1}\left((1 + N_L N_S)N_S \bar{U} + N_L \bar{Y}\right) \\
&\le \bar{X}_2 \stackrel{\text{def}}{=} \frac{\kappa_2(1 + N_L N_S)N_S \bar{U} + \kappa_2(1 + \gamma_2)N_L \bar{Y}}{\gamma_2}.
\end{aligned}
$$

$\square$

At the end of this section, we set the value of $T_w$ so that the confidence set $\mathcal{C}$ is accurate enough to stabilize the LQG system:

$$T_w \stackrel{\text{def}}{=} \max(T_A, T_B, T_C, T_L, T'_A, T'_B, T_{g0}, T_{g1}, T_{g2}, T_M) = \tilde{O}(\bar{\mathcal{K}}^2). \tag{56}$$

## D  PROOF OF LEMMA 3

*Proof of Lemma 3.* The key observation to prove the lemma is that for any $\Theta \in \mathscr{E}$, the difference of $H$-step optimal cost between the infinite-horizon setting of $\Theta$ and finite-horizon setting of $\Theta$ is bounded:
$$|V^\star(\Theta) - (H + 1)J^\star(\Theta)| \le D_h,$$
where $D_h$ is independent of $H$. We now prove this statement.

We first recall the optimal control of the finite horizon LQG problem. The cost for any policy $\pi$ in a $H$-step finite horizon LQG problem is defined as

$$\mathbb{E}_\pi\left[\sum_{h=0}^{H} y_h^\top Q y_h + u_h^\top R u_h\right],$$

where the initial state $x_0 \sim \mathcal{N}(0, I)$.

The optimal control depends on the belief state $\mathbb{E}[x_h|\mathcal{H}_h]$, where $\mathcal{H}_h$ is the history observations and actions up to time step $h$. It is well known that $x_h$ is a Gaussian variable, whose mean and covariance can be estimated by the Kalman filter.

Define the initial covariance estimate $\Sigma_{0|-1} \overset{\text{def}}{=} I$, then Kalman filter gives

$$\Sigma_{h+1|h} = A\Sigma_{h|h-1}A^\top - A\Sigma_{h|h-1}C^\top(C\Sigma_{h|h-1}C^\top + I)^{-1}C\Sigma_{h|h-1}A^\top + I, \qquad (57)$$

and

$$\Sigma_{h|h} = (I - L_hC)\Sigma_{h|h-1},$$

where

$$L_h = \Sigma_{h|h-1}C^\top \left(C\Sigma_{h|h-1}C^\top + I\right)^{-1}.$$

The optimal control is linear in the belief state $\mathbb{E}[x_h|\mathcal{H}_h]$, with an optimal control matrix $K_h$. This control matrix in turn depends on a Riccati difference equation as follows. Define $P_H \overset{\text{def}}{=} C^\top QC$, the Riccati difference equation for $P_h$ takes the form:

$$P_h = A^\top P_{h+1}A + C^\top QC - A^\top P_{h+1}B(R + B^\top P_{h+1}B)^{-1}B^\top P_{h+1}A. \qquad (58)$$

The optimal control matrix is $K_h = (R + B^\top P_h B)^{-1}B^\top P_h A$. Then the optimal cost of the finite horizon LQG problem $V^\star(\Theta)$ equals

$$V^\star(\Theta) = \sum_{h=0}^{H} \text{trace}\left(P_h(\Sigma_{h|h-1} - \Sigma_{h|h})\right) + \sum_{h=0}^{H} \text{trace}\left(C^\top QC\Sigma_{h|h}\right). \qquad (59)$$

For the infinite-horizon view of the same LQG instance $\Theta$, we denote the stable solution to two DAREs in (57) and (58) by $\Sigma$ and $P$ (i.e., we use $\Sigma$, $P$ and $L$ to represent $\Sigma(\Theta)$, $P(\Theta)$ and $L(\Theta)$), then the optimal cost $J^\star(\Theta)$ equals

$$J^\star(\Theta) = \text{trace}\left(PLC\Sigma\right) + \text{trace}\left(C^\top QC(I - LC)\Sigma\right). \qquad (60)$$

Now let us check the difference between $(H+1)J^\star(\Theta)$ and $V^\star(\Theta)$. The difference can be bounded as

$$V^\star(\Theta) - (H+1)J^\star(\Theta) = \sum_{h=0}^{H}\left(\text{trace}\left(P_hL_hC\Sigma_{h|h-1}\right) - \text{trace}\left(PLC\Sigma\right)\right) \qquad (61)$$

$$+ \sum_{h=0}^{H}\left(\text{trace}\left(C^\top QC(I - L_hC)\Sigma_{h|h-1}\right) - \text{trace}\left(C^\top QC(I - LC)\Sigma\right)\right)$$

$$(62)$$

Positive definite matrix $Q$ can be decomposed as $Q = \Gamma^\top\Gamma$, where $\Gamma \in \mathbb{R}^{p\times p}$ is also positive definite. Therefore, for matrix $C^\top QC = (\Gamma C)^\top \Gamma C$, we can show that $(A, \Gamma C)$ is also observable owing to $(A, C)$ observable and $\Gamma$ positive definite. Since $(A, \Gamma C)$ is observable, and $(A, B)$ is controllable, we know that the DARE for $P$ has a unique positive semidefinite solution, and the Riccati difference sequence converges *exponentially fast* to that unique solution (see e.g., Chan et al. (1984, Theorem 4.2), Caines & Mayne (1970, Theorem 2.2), and the references therein). That is, there exists a uniform constant $a_0$ such that

$$\|P - P_h\|_2 \le a_0\gamma_1^{H-h}, \qquad (63)$$

since $\|A - BK\|_2 \le \gamma_1$ for any $\Theta \in \mathscr{E}$.

Similarly, $\Sigma_{h|h-1}$ also converges to $\Sigma$ exponentially fast in that there exists a (instance-dependent) constant $b_0$ (Chan et al., 1984; Lale et al., 2021) such that

$$\left\|\Sigma - \Sigma_{h|h-1}\right\|_2 \le b_0\kappa_2(1 - \gamma_2)^h, \qquad (64)$$

which further implies that $L_h$ also converges to $L$ exponentially fast for some constant $c_0$:

$$\|L - L_h\|_2 \le c_0\kappa_2(1 - \gamma_2)^h, \qquad (65)$$

since $\Sigma_{0|-1} = I$ is positive definite (the techniques of Lemma 3.1 of Lale et al. (2020c) can be used here).

To show the existence of $b_0$, observe that

$$\Sigma_{h|h-1} - \Sigma = (A - ALC)^h (\Sigma_{0|-1} - \Sigma)\phi_h, \qquad (66)$$

where $\phi_h \overset{\text{def}}{=} (A - AL_0C)^\top (A - AL_1C)^\top \cdots (A - AL_{h-1}C)^\top$.

In the proof of Theorem 4.2 of Chan et al. (1984) we have

$$\Sigma_{h|h-1} = \phi_h^\top \Sigma_{0|-1} \phi_h + \text{ non-negative constant matrices.} \tag{67}$$

Theorem 4.1 of Caines & Mayne (1970) shows that for any $\Theta \in \mathscr{E}$ and any $0 \le h \le H$ it holds that

$$\|\Sigma_{h|h-1}\|_2 \le \frac{\kappa_2^2(1 + \kappa_2^2)}{2\gamma_2 - \gamma_2^2} = N_\Sigma, \tag{68}$$

which helps us determine $b_0 = (1 + N_\Sigma)\sqrt{N_\Sigma}$.

The existence of $a_0$ also relies on the state transition matrix (Zhang et al., 2021) $\phi_h' = (A - BK_H)(A - BK_{H-1}) \cdots (A - BK_{h+1})$. As a minimum requirement to tackle the finite horizon LQR control problem, we shall assume $\|\phi_h'\|_2$ is uniformly upper bounded (i.e., the system $\Theta$ is stable) for any $\Theta$ (in fact, people often assume a stronger condition that it is exponentially stable). According to Assumption 2, we can now identify the existence of $a_0$.

As a result, we have

$$|V^\star(\Theta^\star) - (H+1)J^\star(\Theta^\star)| = \left| \sum_{h=0}^{H} \left( \text{trace}\left(P_h L_h C \Sigma_{h|h-1}\right) - \text{trace}\left(PLC\Sigma\right) \right) \right. \tag{69}$$

$$+ \sum_{h=0}^{H} \left( \text{trace}\left(C^\top QC(I - L_hC)\Sigma_{h|h-1}\right) - \text{trace}\left(C^\top QC(I - LC)\Sigma\right) \right) \Bigg| \tag{70}$$

$$\le \sum_{h=0}^{H} \left| \text{trace}\left((P_h - P)L_hC\Sigma_{h|h-1}\right) \right| + \sum_{h=0}^{H} \left| \text{trace}\left(P\left(L_hC\Sigma_{h|h-1} - LC\Sigma\right)\right) \right| \tag{71}$$

$$+ \sum_{h=0}^{H} \left| \text{trace}\left(C^\top QC\left(\Sigma_{h|h-1} - \Sigma + LC\Sigma - L_hC\Sigma_{h|h-1}\right)\right) \right| \tag{72}$$

$$\le \sum_{h=0}^{H} O\left(na_0\gamma_1^{H-h} + n\kappa_2(b_0 + c_0)(1 - \gamma_2)^h\right) \tag{73}$$

$$\le D_h \overset{\text{def}}{=} O\left(\frac{na_0}{1 - \gamma_1} + \frac{n\kappa_2(b_0 + c_0)}{\gamma_2}\right), \tag{74}$$

where the second inequality uses $\text{trace}(X) \le n\|X\|_2$ for any $X \in \mathbb{R}^{n \times n}$.

Now for any (history-dependent) policy $\pi$ and any $\Theta \in \mathscr{E}$,

$$V^\pi(\Theta) - V^\star(\Theta) \le \underbrace{V^\pi(\Theta) - (H+1)J^\star(\Theta)}_{\text{Regret}(\pi;H)} + D_h.$$

Combined with the definition of $\pi_{\text{RT}}$ in Eqn. (7), we have

$$\text{Gap}(\pi_{\text{RT}}) = V^{\pi_{\text{RT}}}(\Theta^\star) - V^\star(\Theta^\star) \le \max_{\Theta \in \mathscr{E}}(V^\pi(\Theta) - V^\star(\Theta)) \le \text{Regret}(\pi; H) + D_h, \tag{75}$$

where the first inequality is by the minimax property of $\pi_{\text{RT}}$ and $\Theta^\star \in \mathscr{E}$.

$\square$

# E    PROOF OF THEOREM 4

For all the conclusions in this section, the bounded state property of LQG-VTR (Lemma 5) is a crucial condition. Let $\mathcal{V}$ be the event that for all $1 \le t \le H$, $\|\hat{x}_{t|t,\tilde{\Theta}_t}\|_2 \le \bar{X}_1$ (so that LQG-VTR does not terminate at Line 9), then by Lemma 5 we know $\mathbb{P}(\mathcal{V}) \ge 1 - \delta$. We provide the full proof of Theorem 4 in this section, but defer the proof of technical lemmas to Appendix F.

### E.1 Bounded $\ell_\infty$ norm of $\mathcal{F}$

To begin with, we show that $\|\mathcal{F}\|_\infty$ is well bounded under event $\mathcal{V}$.

**Lemma 6** (Bounded norm of $\mathcal{F}$). *Suppose event $\mathcal{V}$ happens, then each sample*

$$E_t = (\tau_t, \hat{x}_{t|t,\tilde{\Theta}_k}, \tilde{\Theta}_k, u_t, \hat{x}_{t+1|t+1,\tilde{\Theta}_k}, y_{t+1})$$

*(Line 13 of Algorithm 1) satisfies for any $1 \le t \le H$, $(\tau_t, \hat{x}_{t|t,\tilde{\Theta}_k}, u_t, \tilde{\Theta}_k) \in \mathcal{X}$ (recall that $\mathcal{X}$ is the domain of $f$ defined in Section 4.4).*

*There exists a constant $D$ such that for any $\Theta \in \mathscr{E}$,*

$$\|f_\Theta\|_\infty \overset{\text{def}}{=} \max_{E \in \mathcal{X}} \|f_\Theta(E)\| \le D = O((n+p)\log^2 H),$$

$$\|h_\Theta^\star\|_\infty \le D.$$

### E.2 The Optimistic Confidence Set

Now that $\|\mathcal{F}\|_\infty$ is bounded, we can show the real-world model $\Theta^\star$ is contained in the confidence set with high probability. Note that this lemma requires a more complicated analysis since we are using a *biased* value target regression (Ayoub et al., 2020).

**Lemma 7** (Optimism). *It holds that with probability at least $1 - 2\delta$,*

$$\Theta^\star \in \mathcal{U}_k$$

*for any episode $k \in [\mathcal{K}]$.*

### E.3 The Clipping Error

We also analyze the clipping error introduced by replacing $f'$ with $f$ (see Section 4.4) in the LQG-VTR.

**Lemma 8.** *Suppose event $\mathcal{V}$ happens, there exists constant $\Delta$ such that for any system $\Theta = (A, B, C) \in \mathscr{E}$ and for any input $(\tau^l, x, u, \tilde{\Theta}) \in \mathcal{X}$ at time step $t$,*

$$\Delta_f\left(\tau^t, x, u, \tilde{\Theta}\right) \overset{\text{def}}{=} \left| f'\left(\tau^t, x, u, \tilde{\Theta}\right) - f\left(\tau^l, x, u, \tilde{\Theta}\right) \right| \le \Delta = O\left(\kappa_2(1-\gamma_2)^l(n+p)\log^2 H\right), \tag{76}$$

*where $\tau^t = \{y_0, u_0, y_1, ..., y_t\}$ is the full history, and $\tau^l = \{u_{t-l}, y_{t-l+1}, \cdots, u_{t-1}, y_t\}$ is the clipped history.*

### E.4 Low Switching Property

At last, we observe that the episode switching protocol based on the importance score (Line 14 of Algorithm 1) ensures that the total number of episodes will be only $O(\log H)$.

**Lemma 9** (Low Switching Cost). *Suppose event $\mathcal{V}$ happens and setting $\psi = 4D^2 + 1$, the total number of episodes $\mathcal{K}$ of LQG-VTR(Algorithm 1) is bounded by*

$$\mathcal{K} < \bar{\mathcal{K}},$$

*where $\bar{\mathcal{K}}$ is defined as*

$$\bar{\mathcal{K}} \overset{\text{def}}{=} C_\mathcal{K} \dim_E(\mathcal{F}, 1/H) \log^2(DH) \tag{77}$$

*for some constant $C_\mathcal{K}$.*

*Proof.* This lemma is implied by Lemma 6 and the proof of Chen et al. (2021, Lemma 9).  $\square$

This lemma also implies that the number of episode will NOT reach $\bar{\mathcal{K}}$ since $\mathcal{K} < \bar{\mathcal{K}}$ but not $\mathcal{K} \le \bar{\mathcal{K}}$. This property is important as it guarantees that LQG-VTR switches to a new episode immediately as long as the importance score gets greater than 1.

### E.5 PROOF OF THEOREM 4

We have prepared all the crucial lemmas for the proof of the main theorem till now. Putting everything together, we provide the full proof for the Theorem 4 below.

*Proof of Theorem 4.* Define

$$\widetilde{\mathrm{Regret}}(H) \overset{\mathrm{def}}{=} \sum_{t=T_w+1}^{H} (y_t^\top Q y_t + u_t^\top R u_t - J(\Theta^\star))$$

, then $\mathrm{Regret}(H) = \mathbb{E}[\widetilde{\mathrm{Regret}}(H)] + O(T_w)$. We assume event $\mathcal{V}$ happens and the optimistic property of $\mathcal{U}_k$ (Lemma 7) holds for now, leaving the failure of these two events at the end of the proof. As a consequence of $\mathcal{V}$, the algorithm will not terminate at Line 9.

As we know the regret incurred in the model selection stage is $O(T_w) = O(\bar{\mathcal{K}}^2)$ is only logarithmic in $H$, it cannot be the dominating term of the regret. Denote the full history at time step $t$ as $\tau^t = \{y_0, u_0, y_1, ..., u_{t-1}, y_t\}$, we decompose the regret as

$$\widetilde{\mathrm{Regret}}(H) = \sum_{t=T_w+1}^{H} \left( y_t^\top Q y_t + u_t^\top R u_t - J^\star(\Theta^\star) \right) \tag{78}$$

$$\leq \sum_{t=T_w+1}^{H} \left( y_t^\top Q y_t + u_t^\top R u_t - J^\star(\tilde{\Theta}_k) \right) \quad \text{(by Lemma 7)} \tag{79}$$

$$= \sum_{k=1}^{\mathcal{K}} \sum_{t=S(k)}^{T(k)} \mathbb{E}_{w_t,z_{t+1}} \left[ \hat{x}_{t+1|t+1,\tilde{\Theta}_k}^{u_t \top} \left( \tilde{P} - \tilde{C}^\top Q \tilde{C} \right) \hat{x}_{t+1|t+1,\tilde{\Theta}_k}^{u_t} + y_{t+1,\tilde{\Theta}_k}^{u_t \top} Q y_{t+1,\tilde{\Theta}_k}^{u_t} \right] \tag{80}$$

$$- \hat{x}_{t|t,\tilde{\Theta}_k}^\top \left( \tilde{P} - \tilde{C}^\top Q \tilde{C} \right) \hat{x}_{t|t,\tilde{\Theta}_k} - y_t^\top Q y_t \quad \text{(by the Bellman equation of system } \tilde{\Theta}_k) \tag{81}$$

$$= \sum_{k=1}^{\mathcal{K}} \sum_{t=S(k)}^{T(k)} \mathbb{E}_{w_t,z_{t+1}} \left[ \hat{x}_{t+1|t+1,\tilde{\Theta}_k}^{u_t \top} \left( \tilde{P} - \tilde{C}^\top Q \tilde{C} \right) \hat{x}_{t+1|t+1,\tilde{\Theta}_k}^{u_t} + y_{t+1,\tilde{\Theta}_k}^{u_t \top} Q y_{t+1,\tilde{\Theta}_k}^{u_t} \right] \tag{82}$$

$$- f_{\Theta^\star}' \left( \tau^t, \hat{x}_{t|t,\tilde{\Theta}_k}, u_t, \tilde{\Theta}_k \right) \tag{83}$$

$$+ f_{\Theta^\star}' \left( \tau^t, \hat{x}_{t|t,\tilde{\Theta}_k}, u_t, \tilde{\Theta}_k \right) - \hat{x}_{t|t,\tilde{\Theta}_k}^\top \left( \tilde{P} - \tilde{C}^\top Q \tilde{C} \right) \hat{x}_{t|t,\tilde{\Theta}_k} - y_t^\top Q y_t. \tag{84}$$

Here we denote $\tilde{\Theta}_k = (\tilde{A}, \tilde{B}, \tilde{C})$, and $\tilde{P} = P(\tilde{\Theta}_k)$. For any $k \in [\mathcal{K}]$, the last term (84) of the regret decomposition can be bounded as

$$\sum_{t=S(k)}^{T(k)} f_{\Theta^\star}' \left( \tau^t, \hat{x}_{t|t,\tilde{\Theta}_k}, u_t, \tilde{\Theta}_k \right) - \hat{x}_{t|t,\tilde{\Theta}_k}^\top \left( \tilde{P} - \tilde{C}^\top Q \tilde{C} \right) \hat{x}_{t|t,\tilde{\Theta}_k} - y_t^\top Q y_t \tag{85}$$

$$= \sum_{t=S(k)}^{T(k)-1} f_{\Theta^\star}' \left( \tau^t, \hat{x}_{t|t,\tilde{\Theta}_k}, u_t, \tilde{\Theta}_k \right) - \hat{x}_{t+1|t+1,\tilde{\Theta}_k}^\top \left( \tilde{P} - \tilde{C}^\top Q \tilde{C} \right) \hat{x}_{t+1|t+1,\tilde{\Theta}_k} - y_{t+1}^\top Q y_{t+1} \tag{86}$$

$$+ f_{\Theta^\star}' \left( \tau^{T(k)}, \hat{x}_{T(k)|T(k),\tilde{\Theta}_k}, u_{T(k)}, \tilde{\Theta}_k \right) - \hat{x}_{S(k)|S(k),\tilde{\Theta}_k}^\top \left( \tilde{P} - \tilde{C}^\top Q \tilde{C} \right) \hat{x}_{S(k)|S(k),\tilde{\Theta}_k} - y_{S(k)}^\top Q y_{S(k)}. \tag{87}$$

Observe that Eqn. (86) is a martingale difference sequence, and Eqn. (87) is bounded by $2D$ by Lemma 6. Summing over $k$ we have with probability $1 - \delta$,

$$\sum_{k=1}^{\mathcal{K}} \sum_{t=S(k)}^{T(k)} f_{\Theta^\star}' \left( \tau^t, \hat{x}_{t|t,\tilde{\Theta}_k}, u_t, \tilde{\Theta}_k \right) - \hat{x}_{t|t,\tilde{\Theta}_k}^\top \left( \tilde{P} - \tilde{C}^\top Q \tilde{C} \right) \hat{x}_{t|t,\tilde{\Theta}_k} - y_t^\top Q y_t \tag{88}$$

$$\leq D\sqrt{H \log(1/\delta)} + 2D\mathcal{K}. \tag{89}$$

Now we focus on Eqn. (82) and (83). Observe that Eqn. (82) can be written as

$$\mathbb{E}_{w_t, z_{t+1}} \left[ \hat{x}^{u_t \top}_{t+1|t+1, \tilde{\Theta}_k} \left( \tilde{P} - \tilde{C}^\top Q \tilde{C} \right) \hat{x}^{u_t}_{t+1|t+1, \tilde{\Theta}_k} + y^{u_t \top}_{t+1, \tilde{\Theta}_k} Q y^{u_t}_{t+1, \tilde{\Theta}_k} \right] \tag{90}$$

$$= f'_{\tilde{\Theta}_k} \left( \tau^t, \hat{x}_{t|t, \tilde{\Theta}_k}, u_t, \tilde{\Theta}_k \right). \tag{91}$$

Therefore, Eqn. (82) and (83) becomes

$$\sum_{k=1}^{\mathcal{K}} \sum_{t=S(k)}^{T(k)} f'_{\tilde{\Theta}_k} \left( \tau^t, \hat{x}_{t|t, \tilde{\Theta}_k}, u_t, \tilde{\Theta}_k \right) - f'_{\Theta^\star} \left( \tau^t, \hat{x}_{t|t, \tilde{\Theta}_k}, u_t, \tilde{\Theta}_k \right). \tag{92}$$

Consider any episode $1 \leq k \leq \mathcal{K}$, define $\mathscr{Z}_k$ to be the dataset used for the regression at the end of episode $k$ ($\mathscr{Z}_0 = \emptyset$). The construction of confidence set $\mathcal{U}_k$ (Eqn. (10)) shows that

$$\left\| f_{\tilde{\Theta}_k} - f_{\Theta^\star} \right\|^2_{\mathscr{Z}_{k-1}} \leq 2\beta. \tag{93}$$

By the definition of importance score (Line 14 of Algorithm 1), we know for any $S(k) \leq t \leq T(k)$,

$$\sum_{s=S(k)}^{t} \left( f_{\tilde{\Theta}_k} \left( \tau^l_s, \hat{x}_{s|s, \tilde{\Theta}_k}, u_s, \tilde{\Theta}_k \right) - f_{\Theta^\star} \left( \tau^l_s, \hat{x}_{s|s, \tilde{\Theta}_k}, u_s, \tilde{\Theta}_k \right) \right)^2 \leq 2\beta + \psi + 4D^2, \tag{94}$$

where $\tau^l_s = \{ u_{s-l}, y_{s-l+1}, ..., u_{s-1}, y_s \}$ if $l \leq s$, or $\tau^l_s$ denotes the full history at step $s$ if $l > s$.

Summing up Eqn. (93) and (94) implies for any $k, t$

$$\sum_{o=1}^{k} \sum_{s=S(o)}^{\min(t, T(o))} \left( f_{\tilde{\Theta}_k} \left( \tau^l_s, \hat{x}_{s|s, \tilde{\Theta}_o}, u_s, \tilde{\Theta}_o \right) - f_{\Theta^\star} \left( \tau^l_s, \hat{x}_{s|s, \tilde{\Theta}_o}, u_s, \tilde{\Theta}_o \right) \right)^2 \leq 4\beta + \psi + 4D^2. \tag{95}$$

Invoking Jin et al. (2021, Lemma 26) with $\mathcal{G} = \mathcal{F} - \mathcal{F}, g_t = f_{\tilde{\Theta}_k} - f_{\Theta^\star}, \omega = 1/H$, and $\mu_s(\cdot) = \mathbf{1}[\cdot = (\tau^l_s, \hat{x}_{s|s, \tilde{\Theta}_o}, u_s, \tilde{\Theta}_o)]$, we have

$$\sum_{k=1}^{\mathcal{K}} \sum_{t=S(k)}^{T(k)} \left| f_{\tilde{\Theta}_k} \left( \tau^l_t, \hat{x}_{t|t, \tilde{\Theta}_k}, u_t, \tilde{\Theta}_k \right) - f_{\Theta^\star} \left( \tau^l_t, \hat{x}_{t|t, \tilde{\Theta}_k}, u_t, \tilde{\Theta}_k \right) \right| \tag{96}$$

$$\leq O \left( \sqrt{\dim_E \left( \mathcal{F}, 1/H \right) \beta H} + D \min(H, \dim_E \left( \mathcal{F}, 1/H \right)) \right). \tag{97}$$

The clipping error between $f'$ and $f$ can be bounded by Lemma 8:

$$\sum_{k=1}^{\mathcal{K}} \sum_{t=S(k)}^{T(k)} \left| f'_{\tilde{\Theta}_k} \left( \tau^t, \hat{x}_{t|t, \tilde{\Theta}_k}, u_t, \tilde{\Theta}_k \right) - f'_{\Theta^\star} \left( \tau^t, \hat{x}_{t|t, \tilde{\Theta}_k}, u_t, \tilde{\Theta}_k \right) \right| \tag{98}$$

$$\leq \sum_{k=1}^{\mathcal{K}} \sum_{t=S(k)}^{T(k)} \left| f_{\tilde{\Theta}_k} \left( \tau^l_t, \hat{x}_{t|t, \tilde{\Theta}_k}, u_t, \tilde{\Theta}_k \right) - f_{\Theta^\star} \left( \tau^l_t, \hat{x}_{t|t, \tilde{\Theta}_k}, u_t, \tilde{\Theta}_k \right) \right| \tag{99}$$

$$+ \sum_{k=1}^{\mathcal{K}} \sum_{t=S(k)}^{T(k)} \Delta_{f_{\tilde{\Theta}_k}} \left( \tau^t, \hat{x}_{t|t, \tilde{\Theta}_k}, u_t, \tilde{\Theta}_k \right) + \Delta_{f_{\Theta^\star}} \left( \tau^t, \hat{x}_{t|t, \tilde{\Theta}_k}, u_t, \tilde{\Theta}_k \right) \tag{100}$$

$$\leq O \left( \sqrt{\dim_E \left( \mathcal{F}, 1/H \right) \beta H} + \kappa_2 (1 - \gamma_2)^l (n + p) H \log^2 H \right). \tag{101}$$

Therefore, we choose $l = O(\log(Hn + Hp))$ to obtain

$$\sum_{k=1}^{\mathcal{K}} \sum_{t=S(k)}^{T(k)} \left| f'_{\tilde{\Theta}_k} \left( \tau^t, \hat{x}_{t|t, \tilde{\Theta}_k}, u_t, \tilde{\Theta}_k \right) - f'_{\Theta^\star} \left( \tau^t, \hat{x}_{t|t, \tilde{\Theta}_k}, u_t, \tilde{\Theta}_k \right) \right| \leq O \left( \sqrt{\dim_E \left( \mathcal{F}, 1/H \right) \beta H} \right),$$

$$\tag{102}$$

where $\beta = O(D^2 \log(\mathcal{N}(\mathcal{F}, 1/H)/\delta))$ is defined in Eqn. (119).

Plugging it into Eqn. (92) combined with Eqn. (89), we can finally bound the regret as

$$\widetilde{\text{Regret}}(H) \leq O\left(\sqrt{\dim_E\left(\mathcal{F}, 1/H\right)\beta H} + D\sqrt{H\log(1/\delta)} + 2D\mathcal{K}\right) \tag{103}$$

as long as $\mathcal{V}$ happens, the optimistic property $\Theta^\star \in \mathcal{U}_k$ for any episode $k$, and the martingale difference (86) converges. The probability that any of these three events fail is $4\delta$, in which case the $\widetilde{\text{Regret}}(H)$ can be very large.

The tail bound of the Gaussian variables (Abbasi-Yadkori & Szepesvári, 2011) indicates that for any $q > 0$ and $t > 0$,

$$\mathbb{P}\left(\|w_t\|_2 \geq q\right) \leq 2n\exp\left(-\frac{q^2}{2n}\right), \quad \mathbb{P}\left(\|v_t\|_2 \geq q\right) \leq 2p\exp\left(-\frac{q^2}{2p}\right).$$

Therefore, a union bound implies

$$\mathbb{P}\left(\exists t, \max(\|w_t\|_2, \|v_t\|_2) \geq q\right) \leq 2Hn\exp\left(-\frac{q^2}{2n}\right) + 2Hp\exp\left(-\frac{q^2}{2p}\right).$$

Note that as long as $\forall 0 \leq t \leq H, \|w_t\|_2, \|v_t\|_2 \leq q$, we know that

$$\widetilde{\text{Regret}}(H) \leq C_R(q^2 + \bar{U}^2),$$

since $\|u_t\|_2 \leq \bar{U}$ always hold for some instance dependent constant $C_R$ thanks to the terminating condition (Line 9 of LQG-VTR).

Thus we know

$$\mathbb{P}\left(\widetilde{\text{Regret}}(H) > C_R(q^2 + \bar{U}^2)\right) \leq 2Hn\exp\left(-\frac{q^2}{2n}\right) + 2Hp\exp\left(-\frac{q^2}{2p}\right).$$

We set $q = Hnp$ and $\delta = C_R^{-1}(q^2 + \bar{U}^2)^{-1}$ so that

$$\text{Regret}(H) = \mathbb{E}\left[\widetilde{\text{Regret}}(H)\right] + O(T_w) \tag{104}$$

$$\leq O\left(\sqrt{\dim_E\left(\mathcal{F}, 1/H\right)\beta H}\right) + 4C_R\delta(q^2 + \bar{U}^2) + \int_{C_R(q^2+\bar{U}^2)}^{+\infty} \mathbb{P}\left(\widetilde{\text{Regret}}(H) > x\right)\mathrm{d}x \tag{105}$$

$$\leq O\left(\sqrt{\dim_E\left(\mathcal{F}, 1/H\right)\beta H}\right). \tag{106}$$

The integral above is a constant because $\mathbb{P}(\widetilde{\text{Regret}}(H) > x)$ is a exponentially small term. The theorem is finally proved since $\beta = \tilde{O}(D^2\log\mathcal{N}(\mathcal{F}, 1/H))$ and $\|\mathcal{F}\|_\infty \leq D = \tilde{O}(n + p)$. $\qquad\square$

## F  PROOF OF TECHNICAL LEMMAS IN APPENDIX E

### F.1  PROOF OF LEMMA 6

*Proof of Lemma 6.* The first part of the lemma is implied by the definition of $\mathcal{V}$.

By definition of $\mathcal{X}$, we know that for any $(\tau^l, x, u, \tilde{\Theta}) \in \mathcal{X}$ and $\Theta \in \mathscr{E}$,

$$f_\Theta(\tau^l, x, u, \tilde{\Theta}) = \mathbb{E}_{e_t} \left[ x^\top (\tilde{P} - \tilde{C}^\top Q \tilde{C}) x + y_{t+1,\Theta}^{c\top} Q y_{t+1,\Theta}^c \right] \quad \text{(by Eqn. (16)}) \tag{107}$$

$$= \mathbb{E}_{e_t} \left[ \left( \left( I - \tilde{L}\tilde{C} \right) \left( \tilde{A}x + \tilde{B}u \right) + \tilde{L} \left( CA\hat{x}_{t|t,\Theta}^c + CBu + e_t \right) \right)^\top (\tilde{P} - \tilde{C}^\top Q \tilde{C}) \times \tag{108}$$

$$\left( \left( I - \tilde{L}\tilde{C} \right) \left( \tilde{A}x + \tilde{B}u \right) + \tilde{L} \left( CA\hat{x}_{t|t,\Theta}^c + CBu + e_t \right) \right) \right] \tag{109}$$

$$+ \mathbb{E}_{e_t} \left[ \left( CA\hat{x}_{t|t,\Theta}^c + CBu + e_t \right)^\top Q \left( CA\hat{x}_{t|t,\Theta}^c + CBu + e_t \right) \right] \tag{110}$$

$$= \left( \tilde{L}CA\hat{x}_{t|t,\Theta}^c + \tilde{L}CBu + \left( I - \tilde{L}\tilde{C} \right) \left( \tilde{A}x + \tilde{B}u \right) \right)^\top (\tilde{P} - \tilde{C}^\top Q \tilde{C}) \times \tag{111}$$

$$\left( \tilde{L}CA\hat{x}_{t|t,\Theta}^c + \tilde{L}CBu + \left( I - \tilde{L}\tilde{C} \right) \left( \tilde{A}x + \tilde{B}u \right) \right) \tag{112}$$

$$+ \left( CA\hat{x}_{t|t,\Theta}^c + CBu \right)^\top Q \left( CA\hat{x}_{t|t,\Theta}^c + CBu \right) \tag{113}$$

$$+ \text{trace} \left( \tilde{L}^\top (\tilde{P} - \tilde{C}^\top Q \tilde{C}) \tilde{L} \left( C\Sigma C^\top + I \right) \right) + \text{trace} \left( Q \left( C\Sigma C^\top + I \right) \right), \tag{114}$$

where $\tilde{P} = P(\tilde{\Theta})$, $\tilde{L} = L(\tilde{\Theta})$, and $e_t = Cw_t + CA(x_t - \hat{x}_{t|t,\Theta}) + z_{t+1}$ is the innovation noise such that $e_t \sim \mathcal{N}(0, C\Sigma(\Theta)C^\top + I)$ (Zheng et al., 2021; Lale et al., 2021). .

Note that $\hat{x}_{t|t,\Theta}^c = \hat{x}_{t|t,\Theta} - (A - LCA)^l \hat{x}_{t-l|t-l,\Theta}$, so $\|\hat{x}_{t|t,\Theta}^c\|_2 \le (1 + \kappa_2(1-\gamma_2)^l)\bar{X}_2$. Therefore, each term in Eqn. (111) - (114) is bounded in at most $O((\sqrt{n} + \sqrt{p})\log H)$ besides $\|\hat{x}_{t|t,\Theta}^c\|_2 = O((\sqrt{n} + \sqrt{p})\log H)$. We can finally find a constant $D_1$ such that $\|f_\Theta\|_\infty \le D_1 = O((n + p)\log^2 H)$ for any $\Theta \in \mathscr{E}$.

By the definition of optimal bias function in (4), we have

$$\left| h_\Theta^\star \left( \hat{x}_{t|t,\Theta}, y_t \right) \right| = \left| \hat{x}_{t|t,\Theta}^\top (P(\Theta) - C^\top QC) \hat{x}_{t|t,\Theta} + y_t^\top Q y_t \right| \tag{115}$$

$$\le D_2 \overset{\text{def}}{=} (N_P + N_S^2 N_U) \bar{X}_2^2 + N_U \bar{Y}^2. \tag{116}$$

It suffices to choose $D \overset{\text{def}}{=} \max(D_1, D_2) = O((n + p)\log^2 H)$. $\qquad \square$

## F.2 PROOF OF LEMMA 7

*Proof of Lemma 7.* As a starting point, $\Theta^\star \in \mathcal{U}_1$ holds with probability $1 - \delta/4$ due to the warm up procedure (see Appendix C). We assume event $\mathcal{V}$ happens for now, and discuss the failure of $\mathcal{V}$ at the end of the proof.

For episode $1 \le k \le \mathcal{K}$ and step $t$ in the episode, define $X_t \overset{\text{def}}{=} (\tau^t, \hat{x}_{t|t,\tilde{\Theta}_k}, u_t, \tilde{\Theta}_k), d_t \overset{\text{def}}{=} f_{\Theta^\star}'(X_t) - f_\Theta(X_t), Y_t \overset{\text{def}}{=} \hat{x}_{t+1|t+1,\tilde{\Theta}_k}^\top (\tilde{P} - \tilde{C}^\top Q \tilde{C}) \hat{x}_{t+1|t+1,\tilde{\Theta}_k} + y_{t+1}^\top Q y_{t+1}$ for $\tilde{P} = P(\tilde{\Theta}_k), \tilde{C} = C(\tilde{\Theta}_k)$ (note that $f_{\Theta^\star}$ only depends on the $l$-step nearest history $\tau^l$ instead of the full history $\tau^t$).

Let $\mathcal{F}_{t-1}$ be the filtration generated by $(X_0, Y_0, X_1, Y_1, ..., X_{t-1}, Y_{t-1}, X_t)$, then we know that $f_{\Theta^\star}'(X_t) = \mathbb{E}[Y_t \mid \mathcal{F}_{t-1}]$ by definition. Define $Z_t \overset{\text{def}}{=} Y_t - f_{\Theta^\star}'(X_t)$, then $Z_t$ is a $D/2$-subGaussian random variable conditioned on $\mathcal{F}_{t-1}$ by Lemma 6, and it is $\mathcal{F}_t$ measurable (hence $\mathcal{F}$-adapted). It holds that for the dataset $\mathcal{Z}$ at the end of episode $k$

$$\|Y - f\|_{\mathcal{Z}}^2 - \|Y - f_*\|_{\mathcal{Z}}^2 = \|f_* - f\|_{\mathcal{Z}}^2 + 2 \langle Z + d, f_* - f \rangle_{\mathcal{Z}},$$

where $\langle x, y \rangle_{\mathcal{Z}} \overset{\text{def}}{=} \sum_{e \in \mathcal{Z}} x(e)y(e), \|x - y\|_{\mathcal{Z}}^2 \overset{\text{def}}{=} \langle x - y, x - y \rangle_{\mathcal{Z}}, f_* \overset{\text{def}}{=} f_{\Theta^\star}$.

Rearranging the terms gives

$$\frac{1}{2} \|f_* - f\|_{\mathcal{Z}}^2 = \|Y - f\|_{\mathcal{Z}}^2 - \|Y - f_*\|_{\mathcal{Z}}^2 + E(f),$$

for
$$E(f) \overset{\text{def}}{=} -\frac{1}{2} \|f_* - f\|_{\mathcal{Z}}^2 + 2 \langle Z + d, f - f_* \rangle_{\mathcal{Z}}.$$

Recall that $\hat{f} \overset{\text{def}}{=} f_{\hat{\Theta}_{k+1}} = \arg\min_{\Theta \in \mathcal{U}_1} \|f_\Theta - Y\|_{\mathcal{Z}}^2$ and $f_* \in \mathcal{U}_1$, we have $\|\hat{f} - Y\|_{\mathcal{Z}}^2 \le \|f_* - Y\|_{\mathcal{Z}}^2$. Thus
$$\frac{1}{2} \left\| f_* - \hat{f} \right\|_{\mathcal{Z}}^2 \le E(\hat{f}).$$

In order to show that $\hat{f}$ is close to $f_*$, it suffices to bound $E(\hat{f})$. For some fixed $\alpha > 0$, let $G(\alpha)$ be an $\alpha$-cover of $\mathcal{F}$ in terms of $\| \cdot \|_\infty$. Let $\bar{f} \overset{\text{def}}{=} \min_{f \in \mathcal{F}} \|\hat{f} - f\|_{\mathcal{Z}}$, then
$$E\left(\hat{f}\right) = E\left(\hat{f}\right) - E\left(\bar{f}\right) + E\left(\bar{f}\right) \le E\left(\hat{f}\right) - E\left(\bar{f}\right) + \max_{f \in \mathcal{G}(\alpha)} E(f).$$

We now start to bound these three terms. For any fixed $f \in \mathcal{F}$, we know that $2 \langle Z, f - f_* \rangle_{\mathcal{Z}}$ is $D\|f - f_*\|_{\mathcal{Z}}$-subGaussian. Then it holds with probability $1 - 3\delta/4$ for any episode $k$ and $\lambda > 0$ that
$$E(f) \le -\frac{1}{2} \|f_* - f\|_{\mathcal{Z}}^2 + \frac{1}{\lambda} \log\left(\frac{4}{3\delta}\right) + \lambda \frac{D^2 \|f - f_*\|_{\mathcal{Z}}^2}{2} + 2 \langle d, f - f_* \rangle_{\mathcal{Z}}.$$

Choosing $\lambda = 1/D^2$, we get
$$E(f) \le D^2 \log\left(\frac{4}{3\delta}\right) + 2 \langle d, f - f_* \rangle_{\mathcal{Z}} \le D^2 \log\left(\frac{4}{3\delta}\right) + 2\Delta D H, \tag{117}$$

since $\langle d, f - f_* \rangle_{\mathcal{Z}} \le \|d\|_{\mathcal{Z}} \|f - f_*\|_{\mathcal{Z}} \le \Delta\sqrt{H} \cdot D\sqrt{H} = \Delta D H$.

By a union bound argument we know with probability $1 - 3\delta/4$, the term $\max_{f \in \mathcal{G}(\alpha)} E(f)$ is bounded by $D^2 \log(4|G(\alpha)|/3\delta) + 2\Delta D H$.

For $E(\hat{f}) - E(\bar{f})$, we have
$$\begin{aligned}
E\left(\hat{f}\right) - E\left(\bar{f}\right) &= \frac{1}{2} \|\bar{f} - f_*\|_{\mathcal{Z}}^2 - \frac{1}{2} \left\|\hat{f} - f_*\right\|_{\mathcal{Z}}^2 + 2\left\langle Z + d, \hat{f}_t - \bar{f}\right\rangle_{\mathcal{Z}} \\
&\le \frac{1}{2}\left(\left\langle \bar{f} - \hat{f}, \bar{f} + \hat{f} + 2f_* \right\rangle_{\mathcal{Z}}\right) + 2\left(\|Z\|_{\mathcal{Z}} + \|d\|_{\mathcal{Z}}\right)\left\|\hat{f} - \bar{f}\right\|_{\mathcal{Z}}.
\end{aligned}$$

Note that by definition of $\bar{f}$ it holds that
$$\left\|\bar{f} - \hat{f}\right\|_{\mathcal{Z}} \le \alpha\sqrt{H}.$$

Together with $\|\bar{f}\|_{\mathcal{Z}}, \|\hat{f}\|_{\mathcal{Z}}, \|f_*\|_{\mathcal{Z}} \le D\sqrt{H}$, we bound $E(\hat{f}) - E(\bar{f})$ as
$$\begin{aligned}
E\left(\hat{f}\right) - E\left(\bar{f}\right) &\le 2D\alpha H + \left(2\Delta\sqrt{H} + D\sqrt{2H \log\left(3H(H+1)/\delta\right)}\right) \cdot \alpha\sqrt{H} \\
&\le 2\left(D + \Delta\right)\alpha H + \alpha H \cdot D\sqrt{2\log\left(3H(H+1)/\delta\right)}.
\end{aligned}$$

Here we take advantage of the $D/2$-subGaussian property of $z_t$ in that with probability $1 - 3\delta/4$ for any episode $k$,
$$\|Z\|_{\mathcal{Z}} \le \frac{D}{2}\sqrt{2|\mathcal{Z}| \log\left(2|\mathcal{Z}|(|\mathcal{Z}| + 1)/\delta\right)} \le \frac{D}{2}\sqrt{2H \log\left(3H(H+1)/\delta\right)}. \tag{118}$$

Merging Eqn. (117) and (118) with another union bound, we get that with probability $1 - \delta$ for any episode $k$,
$$\left\|f_* - \hat{f}\right\|_{\mathcal{Z}}^2 \le 2D^2 \log\left(2N_\alpha/\delta\right) + 4\Delta D H + 2\alpha H\left(2(D + \Delta) + D\sqrt{2\log\left(6H(H+1)/\delta\right)}\right),$$

where $N_\alpha$ is the $(\alpha, \|\cdot\|_\infty)$− covering number of $\mathcal{F}$.

Finally, we set $\alpha = 1/H$ and
$$\beta = 2D^2 \log\left(2\mathcal{N}(\mathcal{F}, 1/H)\right) + 4\Delta D H + 4(D + \Delta) + 4D\sqrt{2\log(4H(H+1)/\delta)}. \tag{119}$$

It holds that $\Theta^\star \in \mathcal{U}_k$ for any $1 \le k \le \mathcal{K}$ with probability $1 - \delta/4 - 3\delta/4 = 1 - \delta$ as long as $\mathcal{V}$ happens. Since $\mathcal{V}$ happens with probability $1 - \delta$, we know the optimistic property holds with probability $1 - 2\delta$. $\qquad\square$

### F.3 PROOF OF LEMMA 8

*Proof of Lemma 8.* Define $e^c \overset{\text{def}}{=} (A - LCA)^l \hat{x}_{t-l|t-l,\Theta}$, then $\hat{x}_{t|t,\Theta} = \hat{x}^c_{t|t,\Theta} + e^c$. Further, we know $\|e^c\|_2 \leq \kappa_2 (1 - \gamma_2)^l \bar{X}_2$ under event $\mathcal{V}$ and Assumption 3. By the decomposition rule for $f$ and $f'$ (Eqn. (111) - (114)), we know

$$f'_\Theta(\tau^t, x, u, \tilde{\Theta}) - f_\Theta(\tau^l, x, u, \tilde{\Theta}) \tag{120}$$

$$= 2e^{c\top} A^\top C^\top \tilde{L}^\top (\tilde{P} - \tilde{C}^\top Q \tilde{C}) \left( \tilde{L} C A \hat{x}^c_{t|t,\Theta} + \tilde{L} C B u + \left( I - \tilde{L}\tilde{C} \right) \left( \tilde{A}x + \tilde{B}u \right) \right) \tag{121}$$

$$+ e^{c\top} A^\top C^\top \tilde{L}^\top (\tilde{P} - \tilde{C}^\top Q \tilde{C}) \tilde{L} C A e^c \tag{122}$$

$$+ 2e^{c\top} A^\top C^\top Q \left( C A \hat{x}^c_{t|t,\Theta} + C B u \right) + e^{c\top} A^\top C^\top Q C A e^c. \tag{123}$$

Note that $\bar{X}_2 = O((\sqrt{n} + \sqrt{p}) \log H)$, thereby there exists a constant $\Delta$ such that

$$\left| f'_\Theta(\tau^t, x, u, \tilde{\Theta}) - f_\Theta(\tau^l, x, u, \tilde{\Theta}) \right| \leq \Delta = O \left( \kappa_2 (1 - \gamma_2)^l (n + p) \log^2 H \right). \tag{124}$$

$$\square$$

## G THE INTRINSIC MODEL COMPLEXITY $\delta_{\mathscr{E}}$

In this section, we show how large the intrinsic model complexity $\delta_{\mathscr{E}}$ will be for different simulator classes $\mathscr{E}$. An important message is that $\delta_{\mathscr{E}}$ does NOT have any *polynomial* dependency on $H$.

### G.1 GENERAL SIMULATOR CLASS

Without further structures, we can show that $\delta_{\mathscr{E}}$ is bounded by

$$\delta_{\mathscr{E}} = \tilde{O} \left( np^2(n + m + p)(n + p)^2(m + p)^2 \right) \tag{125}$$

due to Proposition 2, 3 and the fact that $\|\mathcal{F}\|_\infty \leq D = \tilde{O}(n + p)$.

**Proposition 2.** *Under Assumption 1, 2, 3, the $1/H$-Eluder dimension of $\mathcal{F}$ is bounded by*

$$\dim_E (\mathcal{F}, 1/H) = \tilde{O} \left( p^4 + p^3 m + p^2 m^2 \right). \tag{126}$$

*Proof.* The bound on Eluder dimension mainly comes from the fact that $f_\Theta(\tau^l, x, u, \tilde{\Theta})$ can be regarded a *linear* function between features of $\Theta$ and features of $\tilde{\Theta}$. We use the "feature" of a system $\Theta$ to represent a quantity that *only depends on* $\Theta$ (independent from $\tilde{\Theta}$) and vice versa. It is well-known that the linear function class has bounded Eluder dimension. We formalize this idea below.

In the proof of Lemma 8, we know that for any $(\tau^l, x, u, \tilde{\Theta}) \in \mathcal{X}$ and $\Theta \in \mathscr{E}$,

$$f_\Theta(\tau^l, x, u, \tilde{\Theta}) = \left( \tilde{L} C A \hat{x}^c_{t|t,\Theta} + \tilde{L} C B u + \left( I - \tilde{L}\tilde{C} \right) \left( \tilde{A}x + \tilde{B}u \right) \right)^\top (\tilde{P} - \tilde{C}^\top Q \tilde{C}) \times \tag{127}$$

$$\left( \tilde{L} C A \hat{x}^c_{t|t,\Theta} + \tilde{L} C B u + \left( I - \tilde{L}\tilde{C} \right) \left( \tilde{A}x + \tilde{B}u \right) \right) \tag{128}$$

$$+ \left( C A \hat{x}^c_{t|t,\Theta} + C B u \right)^\top Q \left( C A \hat{x}^c_{t|t,\Theta} + C B u \right) \tag{129}$$

$$+ \text{trace} \left( \tilde{L}^\top (\tilde{P} - \tilde{C}^\top Q \tilde{C}) \tilde{L} \left( C \Sigma C^\top + I \right) \right) + \text{trace} \left( Q \left( C \Sigma C^\top + I \right) \right), \tag{130}$$

Denote the input as $E \overset{\text{def}}{=} (\tau^l, x, u, \tilde{\Theta}) \in \mathcal{X}$, and define three feature mappings $\zeta : \mathcal{X} \to \mathbb{R}^{n \times n}$, $\phi : \mathcal{X} \to \mathbb{R}^n$ and $\Phi : \mathcal{X} \to \mathbb{R}^{p \times p}$ such that

$$\zeta(E) = \tilde{P} - \tilde{C}^\top Q \tilde{C}, \tag{131}$$

$$\phi(E) = \left( I - \tilde{L}\tilde{C} \right) \left( \tilde{A}x + \tilde{B}u \right), \tag{132}$$

$$\Phi(E) = \tilde{L}^\top \zeta(E) \tilde{L} + Q. \tag{133}$$

Then we can move on to further write $f$ as

$$f_\Theta(E) = \left( \tilde{L}CA\hat{x}^c_{t|t,\Theta} + \tilde{L}CBu + \phi(E) \right)^\top \zeta(E) \left( \tilde{L}CA\hat{x}^c_{t|t,\Theta} + \tilde{L}CBu + \phi(E) \right) \tag{134}$$

$$+ \left( CA\hat{x}^c_{t|t,\Theta} + CBu \right)^\top Q \left( CA\hat{x}^c_{t|t,\Theta} + CBu \right) + \mathrm{trace}\left( \Phi(E) \left( C\Sigma C^\top + I \right) \right). \tag{135}$$

By definition,

$$\hat{x}^c_{t|t,\Theta} = \sum_{s=1}^{l} (A - LCA)^{s-1} \left( (I - LC)Bu_{t-s} + Ly_{t-s+1} \right). \tag{136}$$

Define two series of feature mappings $\varphi^s_u, \varphi^s_y$ for $1 \le s \le l$ on $\mathscr{E}$ to be $\varphi^s_u : \mathscr{E} \to \mathbb{R}^{p\times m}$ and $\varphi^s_y : \mathscr{E} \to \mathbb{R}^{p\times p}$ such that

$$\begin{aligned} \varphi^s_u(\Theta) &= CA\left( A - LCA \right)^{s-1} (I - LC)B \\ \varphi^s_y(\Theta) &= CA\left( A - LCA \right)^{s-1} L \end{aligned} \tag{137}$$

Consider the terms in Eqn. (134) and (135, each term can be written as the *inner product* of a feature of $\Theta$ and a feature of $E$. For example, $\mathrm{trace}\left( \Phi(E)(C\Sigma C^\top + I) \right)$ is the inner product of $\Phi(E)$ (feature of $E$) and $C\Sigma C^\top + I$ (feature of $\Theta$, since it only depends on $\Theta$). As a more complicated example, we check $(\tilde{L}CA\hat{x}^c_{t|t,\Theta})^\top \zeta(E)(\tilde{L}CA\hat{x}^c_{t|t,\Theta})$.

$$\left( \tilde{L}CA\hat{x}^c_{t|t,\Theta} \right)^\top \zeta(E)\tilde{L}CA\hat{x}^c_{t|t,\Theta} \tag{138}$$

$$= \left( \sum_{s=1}^{l} \varphi^s_u(\Theta)u_{t-s} + \varphi^s_y(\Theta)y_{t-s+1} \right)^\top \tilde{L}^\top \zeta(E)\tilde{L} \left( \sum_{s=1}^{l} \varphi^s_u(\Theta)u_{t-s} + \varphi^s_y(\Theta)y_{t-s+1} \right). \tag{139}$$

Note that $\varphi^s_u(\Theta)$ and $\varphi^s_y(\Theta)$ only depends on $\Theta$ (so they can be regarded as a feature of $\Theta$), where $u_{t-s}, y_{t-s+1}, \tilde{L}^\top \zeta(E)\tilde{L}$ only depends on $E$. Consider any $1 \le s_1, s_2 \le l$ in the summation, we pick the term below as an example:

$$(\varphi^{s_1}_u(\Theta)u_{t-s_1})^\top \tilde{L}^\top \zeta(E)\tilde{L}\varphi^{s_2}_u(\Theta)u_{t-s_2} \tag{140}$$

$$= \left\langle \varphi^{s_2}_u(\Theta) \otimes \varphi^{s_1}_u(\Theta), \mathrm{vec}(\tilde{L}^\top \zeta(E)\tilde{L}) \times \mathrm{vec}(u_{t-s_1}u^\top_{t-s_2})^\top \right\rangle. \tag{141}$$

Here $\otimes$ denotes the Kronecker product between two matrices, $\mathrm{vec}(X)$ for $X \in \mathbb{R}^{a\times b}$ denotes the vectorization of matrix $X$ (i.e., $\mathrm{vec}(X) = [X_{11} \quad X_{21} \quad ... \quad X_{a1} \quad X_{12} \quad ... \quad X_{ab}]^\top \in \mathbb{R}^{ab}$), and $\langle \cdot, \cdot \rangle$ is the inner product. In this way, we decompose the term (140) as the inner product of a $p^2m^2$-dimensional feature of $\Theta$ (i.e., $\varphi^{s_2}_u(\Theta) \otimes \varphi^{s_1}_u(\Theta)$) and a $p^2m^2$-dimensional feature of $E$ (i.e., $\mathrm{vec}(\tilde{L}^\top \zeta(E)\tilde{L}) \times \mathrm{vec}(u_{t-s_1}u^\top_{t-s_2})^\top$).

We then observe that such decomposition is valid for any $1 \le s_1, s_2 \le l$. Therefore, we can decompose each term in Eqn. (134) and (135) as the inner product of features of $\Theta$ and $E$. Gathering all these features as the aggregated feature mapping $\mathrm{fea}_1$ for $\Theta$ and $\mathrm{fea}_2$ for $E$, we have

$$f_\Theta(E) = \langle \mathrm{fea}_1(\Theta), \mathrm{fea}_2(E) \rangle. \tag{142}$$

It is not hard to observe that the dimension of aggregated feature mappings is bounded by

$$\dim(\mathrm{fea}_1(\Theta)) = \dim(\mathrm{fea}_2(E)) = \tilde{O}\left( p^4 + p^3m + p^2m^2 \right) \tag{143}$$

since $l = O(\log(Hn + Hp))$.

Note the $\|\mathrm{fea}_1(\Theta)\|_2, \|\mathrm{fea}_2(E)\|_2$ are both upper bounded by the domain of $E \in \mathcal{X}$ and $\Theta \in \mathscr{E}$, then Proposition 2 of Osband & Van Roy (2014) shows that

$$\dim_E (\mathcal{F}, 1/H) = \tilde{O}\left( p^4 + p^3m + p^2m^2 \right). \tag{144}$$

$\square$

**Proposition 3.** *Under Assumptions 1, 2, and 3, the logarithmic $1/H$-covering number of $\mathcal{F}$ is bounded by*

$$\log \mathcal{N}\left(\mathcal{F}, 1/H\right) = \tilde{O}\left(n^2 + nm + np\right). \tag{145}$$

*Proof.* Under Assumption 2, the spectral norm of the system dynamics of $\Theta = (A, B, C) \in \mathscr{E}$ is bounded by $\|A\|_2, \|B\|_2, \|C\|_2 \leq N_S$. Therefore, we can construct an $\epsilon_0$-net $\mathcal{G}_{\epsilon_0}(\mathscr{E})$ such that for any $\Theta = (A, B, C)$, there exists $\bar{\Theta} = (\bar{A}, \bar{B}, \bar{C}) \in \mathscr{E}$ satisfying $\max(\|A - \bar{A}\|_2, \|B - \bar{B}\|_2, \|C - \bar{C}\|_2) \leq \epsilon_0$. By classic theory, we know $|\mathcal{G}_{\epsilon_0}(\mathscr{E})| \leq O((1 + 2\sqrt{n}N_S/\epsilon_0)^{n^2 + nm + np})$. To see this, observe that $\|\cdot\|_2 \leq \|\cdot\|_F \leq \sqrt{\min(n, m)}\|\cdot\|_2$ for any $n \times m$ matrix, so we can reduce the $\epsilon$-cover with respect to the $l_2$-norm to the $\epsilon$-cover with respect to the Frobenius norm. To show the covering number of $\mathcal{F}$, we check the gap $\|f_\Theta - f_{\bar{\Theta}}\|_\infty$.

By Lemma 3.1 of Lale et al. (2020c), we know that for small enough $\epsilon_0$ there exists instance dependent constants $C_\Sigma$ and $C_L$ such that

$$\left\|\bar{\Sigma} - \Sigma\right\|_2 \leq C_\Sigma \epsilon_0, \quad \left\|\bar{L} - L\right\|_2 \leq C_L \epsilon_0, \tag{146}$$

where $\bar{\Sigma} = \Sigma(\bar{\Theta}), \bar{L} = L(\bar{\Theta})$. The constant $C_\Sigma$ is slightly different from Lemma 3.1 of their paper, because we do not assume $M \overset{\text{def}}{=} A - ALC$ is contractible (i.e., $\|M\|_2 < 1$). Rather we know $M$ is $(\kappa_2, \gamma_2)$-strongly stable. Therefore, the linear mapping $\mathcal{T} \overset{\text{def}}{=} X \to X - MXM^\top$ is invertible with $\|\mathcal{T}^{-1}\|_2 \leq \kappa_2/(2\gamma_2 - \gamma_2^2)$ according to Lemma B.4 of Simchowitz & Foster (2020). As a result, we can set $C_\Sigma$ by replacing the $1/(1 - \nu^2)$ term in the original constant of Lale et al. (2020c, Lemma 3.1) by $\kappa_2/(2\gamma_2 - \gamma_2^2)$.

For any $E = (\tau^l, x, u, \tilde{\Theta}) \in \mathcal{X}$, it is desired to compute the difference $|f_\Theta(E) - f_{\bar{\Theta}}(E)|$. As a first step, we show the difference between $\hat{x}^c_{t|t,\Theta}$ and $\hat{x}^c_{t|t,\bar{\Theta}}$. Note that for any $1 \leq s \leq l$,

$$\left\|(A - LCA)^{s-1} - (\bar{A} - \bar{L}\bar{C}\bar{A})^{s-2}\right\|_2 \leq (s-1)\kappa_2^2(1 - \gamma_2)^{s-1}(1 + N_S^2 + 2N_L N_S)\epsilon_0. \tag{147}$$

This is because

$$\mathcal{A}^{s-1} - \bar{\mathcal{A}}^{s-1} = \sum_{o=0}^{s-2} \mathcal{A}^{s-1-o}\bar{\mathcal{A}}^o - \mathcal{A}^{s-2-o}\bar{\mathcal{A}}^{o+1} = \sum_{o=0}^{s-2} \mathcal{A}^{s-2-o}\left(\mathcal{A} - \bar{\mathcal{A}}\right)\bar{\mathcal{A}}^o \tag{148}$$

for $\mathcal{A} \overset{\text{def}}{=} A - LCA, \bar{\mathcal{A}} \overset{\text{def}}{=} \bar{A} - \bar{L}\bar{C}\bar{A}$, where

$$\left\|\mathcal{A} - \bar{\mathcal{A}}\right\|_2 \leq (1 + N_S^2 + 2N_L N_S)\epsilon_0, \quad \|\mathcal{A}\|_2, \|\bar{\mathcal{A}}\|_2 \leq \kappa_2(1 - \gamma_2).$$

Similarly we can bound $\|(B - LCB) - (\bar{B} - \bar{L}\bar{C}\bar{B})\|_2$. Follow the decomposition rule in Eqn. (136) and observe that $\|u_{t-s}\|_2 \leq \bar{U}, \|y_{t-s+1}\|_2 \leq \bar{Y}$ for $1 \leq s \leq l$, we have

$$\left\|\hat{x}^c_{t|t,\Theta} - \hat{x}^c_{t|t,\bar{\Theta}}\right\|_2 \leq C_x \epsilon_0, \tag{149}$$

where $C_x$ is a instance dependent constant (also depends on $C_\Sigma, C_L$).

By Eqn. (134) and (135), it holds that

$$f_\Theta(E) - f_{\bar{\Theta}}(E) = \text{trace}\left(\Phi(E)\left(C\Sigma C^\top - \bar{C}\bar{\Sigma}\bar{C}^\top\right)\right) + \text{Diff}_1 + \text{Diff}_2, \tag{150}$$

where

$$\text{Diff}_1 = \left(CA\hat{x}^c_{t|t,\Theta} + CBu\right)^\top Q \left(CA\hat{x}^c_{t|t,\Theta} + CBu\right) \tag{151}$$

$$- \left(\bar{C}\bar{A}\hat{x}^c_{t|t,\bar{\Theta}} + \bar{C}\bar{B}u\right)^\top Q \left(\bar{C}\bar{A}\hat{x}^c_{t|t,\bar{\Theta}} + \bar{C}\bar{B}u\right) \tag{152}$$

$$\text{Diff}_2 = \left(\tilde{L}CA\hat{x}^c_{t|t,\Theta} + \tilde{L}CBu\right)^\top \zeta(E) \left(\tilde{L}CA\hat{x}^c_{t|t,\Theta} + \tilde{L}CBu\right) \tag{153}$$

$$- \left(\tilde{L}\bar{C}\bar{C}\hat{x}^c_{t|t,\bar{\Theta}} + \tilde{L}\bar{C}\bar{B}u\right)^\top \zeta(E) \left(\tilde{L}\bar{C}\bar{C}\hat{x}^c_{t|t,\bar{\Theta}} + \tilde{L}\bar{C}\bar{B}u\right). \tag{154}$$

Checking each term in $\text{Diff}_1$ and $\text{Diff}_2$, we conclude that there exists instance dependent constant $C_{d1}, C_{d2}$ such that

$$|\text{Diff}_1| \leq C_{d1}\epsilon_0, \quad |\text{Diff}_2| \leq C_{d2}\epsilon_0.$$

Note that $\|C\Sigma C^\top - \bar{C}\bar{\Sigma}\bar{C}^\top\|_2 \leq (2N_S N_\Sigma + N_S^2)\epsilon_0$ and $\|\Phi(E)\|_2 \leq N_L^2(N_P + N_S^2 N_U) + N_U$, we have

$$|f_\Theta(E) - f_{\bar{\Theta}}(E)| = \left|\text{trace}\left(\Phi(E)\left(C\Sigma C^\top - \bar{C}\bar{\Sigma}\bar{C}^\top\right)\right) + \text{Diff}_1 + \text{Diff}_2\right| \leq C_f\epsilon_0, \quad (155)$$

where

$$C_f \stackrel{\text{def}}{=} p\|\Phi(E)\|_2(2N_S N_\Sigma + N_S^2)\epsilon_0 + C_{d1} + C_{d2}.$$

This means

$$\|f_\Theta - f_{\bar{\Theta}}\|_\infty \leq C_f\epsilon_0.$$

Therefore, the subset $\{f_{\bar{\Theta}} \mid \bar{\Theta} \in \mathcal{G}_{\epsilon_0}(\mathscr{E})\}$ forms a $C_f\epsilon_0$-covering set of $\mathcal{F}$. Finally, if suffices to set $\epsilon_0 = C_f^{-1}H^{-1}$ to induce a $1/H$-covering set of $\mathcal{F}$:

$$\mathcal{N}(\mathcal{F}, 1/H) \leq \left|\mathcal{G}_{C_f^{-1}H^{-1}}(\mathscr{E})\right| = O\left(\left(1 + 2\sqrt{n}N_S C_f H\right)^{n^2 + nm + np}\right), \quad (156)$$

which finishes the proof. $\qquad\square$

## G.2 Low-Rank Simulator Class

In many sim-to-real tasks, there are only a few control parameters that affect the dynamics in the simulator class $\mathscr{E}$ (see e.g., Table 1 of OpenAI et al. (2018)). The number of control parameters is often a constant that is independent of the dimension of the task. This means our simulator class $\mathscr{E}$ is a *low-rank* class in these scenarios:

$$\mathscr{E} = \{\Theta(t_1, t_2, ..., t_k) \mid (t_1, t_2, ..., t_k) \in T\},$$

where $t_1, t_2, ..., t_k$ represent $k$ control parameters, $T$ is the domain of control parameters. The dynamics of simulator $\Theta$ *only* depends on these control parameters $t_1, t_2, ..., t_k$.

It is a common situation that $\Theta(t_1, t_2, ..., t_k)$ is a continuous function of $(t_1, ..., t_k)$. For example, it is a continuous function when the control parameters are physical parameters like friction coefficients, damping coefficients. A simple example is when $\Theta$ is a linear combination of $k$ base simulators:

$$\Theta = (A, B, C) = \sum_{i=1}^{k} t_i \Theta_i, \quad (157)$$

where $\Theta_i = (A_i, B_i, C_i)$ is a fixed base simulator. This can be achieved if we approximate the effect of control parameters with some linear mappings on the states $x_t$. We can set different values of control parameters $t_1, ..., t_k$ to generate new simulators.

In such a continuous low rank class, the log covering number $\mathcal{N}(\mathcal{F}, 1/H)$ will reduce to only $\tilde{O}(k)$ as long as $\Theta$ is continuous with respect to the control parameters. It is straightforward to see this by checking the proof of Proposition 3. Unfortunately, it is unclear whether the Eluder dimension of such a continuous low-rank class will be smaller due to the existence of the Kalman gain matrix $L$. Overall, the intrinsic model complexity $\delta_{\mathscr{E}}$ will decrease from $\tilde{O}(np^2(n+m+p)(n+p)^2(m+p)^2)$ to at most

$$\tilde{O}(kp^2(m+p)^2(n+p)^2)$$

for a small constant $k$.

## G.3 Low-Rank Simulator Class with Real Data

Many works studying sim-to-real transfer also fine-tune the simulated policy with a few real data on the real-world model (Tobin et al., 2017; Rusu et al., 2017). Moreover, the observations are usually the perturbed input states with observation noises introduced by vision inputs (e.g., from the cameras) (Tobin et al., 2017; Peng et al., 2018; OpenAI et al., 2018). This means the observation $y_t$ equals $x_t + z_t$ for random noise $z_t$, without any linear transformation $C$. With these real data

and noisy observations of the state, we can estimate the steady-state covariance matrix $\Sigma(\Theta^\star)$ with random actions since it is independent of the policy! The intrinsic complexity of the simulator set will be further reduced for this low-rank simulator set (e.g., the simulator set defined by Eqn. (157)) combined with real data.

In such low rank simulator classes, we know that $\Sigma(\Theta^\star)$ is fixed as an estimator $\hat{\Sigma}$ computed with noisy observations $y_t = x_t + z_t$. By definition we know $L(\Theta^\star)$ is also fixed as $\hat{L} = \hat{\Sigma}(\hat{\Sigma} + I)^{-1}$. Therefore, the Kalman dynamics matrix $A - LCA = A - \hat{L}A$ belongs to a $k$-dimensional space as $A$ lies in a $k$ dimensional space. As a result, the dimension of feature mappings $\varphi_u^s(t_1, t_2, ..., t_k), \varphi_y^s(t_1, t_2, ..., t_k)$ (Eqn. (137)) reduces to $O(k^2 l^k)$. The Eluder dimension of this low rank simulator class reduces to $\tilde{O}(k^4 l^{2k}) = \tilde{O}(k^4 \log^{2k}(H))$ for a small constant $k$.

Therefore, the intrinsic model complexity $\delta_\mathscr{E}$ for a low-rank simulator class fine-tuned with real data is

$$\tilde{O}((n+p)^2 k^5 \log^{2k}(H)),$$

which implies that the robust adversarial training algorithms are very powerful with a small sim-to-real gap in such simulator classes.

## H    PROOF OF THEOREM 2

*Proof of Theorem 2.* Throughout this proof, we assume $H$ is sufficiently large and omit some (instance-dependent) constants because we focus on the dependency of $H$. Note that LQR is a special case of LQG in the sense that the learner can observe the hidden state in LQR, it suffices to consider the case where $\mathscr{E}$ is a set of LQRs. Since $C = I$ in LQRs, Assumption 3 holds naturally. Let $(A^\star, B^\star)$ be the parameters satisfying Assumptions 1, and 2, and $\mathscr{E} = \{(A, B) : \|A - A^\star\|_\infty \leq \epsilon, \|B - B^\star\|_\infty \leq \epsilon\}$. Choosing $\epsilon \propto H^{-1/2}$ where $H$ is sufficiently large, by the same perturbation analysis in Simchowitz & Foster (2020, Appendix B), we have that all $\Theta \in \mathscr{E}$ satisfy Assumptions 1 and 2. Fix policy $\hat{\pi}$, we have

$$\max_{\Theta \in \mathscr{E}}[V^{\hat{\pi}}(\Theta) - V^*(\Theta)] \geq \min_{\pi} \max_{\Theta \in \mathscr{E}}[V^\pi(\Theta) - V^*(\Theta)]. \tag{158}$$

Since we can treat a history-dependent policy as an algorithm, the lower bound in Simchowitz & Foster (2020, Corollary 1) implies that

$$\min_{\pi} \max_{\Theta \in \mathscr{E}}[V^\pi(\Theta) - V^*(\Theta)] \geq \Omega(\sqrt{H}). \tag{159}$$

Notably, Simchowitz & Foster (2020) only imposes the strongly stable assumption, which is weaker than our assumption, but their proof (Lemma B.7 in Simchowitz & Foster (2020)) still holds under our assumptions. Combining (158) and (159), we conclude the proof of Theorem 2. $\qquad\square$

