# OpenReview forum: "Provable Sim-to-real Transfer in Continuous Domain with Partial Observations"
_ICLR.cc/2023/Conference — ICLR 2023 poster_

### Official Review · Reviewer_86xP · 2022-10-23

**Confidence:** 3
**Correctness:** 4
**Technical Novelty And Significance:** 4
**Empirical Novelty And Significance:** Not applicable
**Recommendation:** 8

**Clarity, Quality, Novelty And Reproducibility:**

- Clarity. See the previous section on weaknesses.

- Novelty. The LQG setting is novel and, as the authors argue, able to capture the challenges in practical applications of Sim-to-real transfer. The proof techniques, especially the history clipping technique, are significantly different from existing works and sufficiently novel.

- Quality. The proofs appear to be correct and the statements seem sound.

- Reproducibility. See previous section on weaknesses. While I am confident that domain experts would have little issue reproducing the proofs, currently the way the work is presented makes it more difficult to reproduce them.

**Strength And Weaknesses:**

Strength
- The paper is theoretically sound and the proofs appear to be correct.
- The paper is well-motivated by a practical need to extend sim-to-real analysis to continuous and partially observable regimes.

Weakness
- Section 4 took many reads to fully appreciate. Particularly, while section 4.1 is reasonable and the reduction technique is standard, as the authors argued, it was not immediately clear why Algorithm 1 (LQG-VTR) is necessary for proving the properties of $\pi_{RT}$ defined in Definition 1. Since the section is the "meat" of the paper, so to speak, providing more intuition and insights into the analytical techniques could make the work easier to digest.
- Some of the parts of the paper are not self contained. For instance, in Appendix C, multiple references to the algorithms and results in Lale et al (2021) are made without providing sufficient discussion on why the results in Lale et al (2021) are useful or intuition on what purposes these references serve, which makes it harder to properly assess the soundness of the paper. A further revision could greatly ameliorate this weakness.

--------- Post Rebuttal Update --------------

Thank the authors for their commitment to making the paper easier to digest. I would like to keep my score as is.

**Summary Of The Paper:**

The paper studies provable sim-to-real transfer in the linear quadratic Gaussian (LQG) setting. The paper proposes a robust learning algorithm for the simulation part and derives provable guarantees without assuming access to real world data. In order to obtain the guarantees, the authors propose a novel history clipping technique that improves the bound significantly.

**Summary Of The Review:**

Overall, I believe the submission offers a novel contribution to the field of sim-to-real transfer. The paper is not perfect and the presentation could certainly be improved. Nevertheless to me the pros outweigh the cons.

---

> ### Author Response · Authors · 2022-11-13
> **Response to Reviewer 86xP**
>
> Thank you for your comments and assessments.
>
> **Regarding the involved technical arguments in Section 4**: Thanks for pointing out this problem. We
> appreciate your suggestion and have revised our technical parts in the main text accordingly. In particular, we
> have restated the reduction lemma in a more direct way so that it is immediate we need to design an algorithm
> with provably low regret to show a small sim-to-real gap of $\pi_{\mathrm{RT}}$. Please check the updated version of the reduction
> lemma.
>
> **Regarding the paper not self-contained**: The main reason we do not provide exact expressions in Appendix
> C is that the exact expressions are too much involved with complicated instance-dependent parameters as in Lale et
> al. (2020). We simplify these complicated terms to make our paper clean. As mentioned in the appendix, the missing
> parts can be derived from Lale et al. (2020) in a direct way. We have provided more details for these derivations in
> the appendix according to your comment, and hopefully these details would address your concerns.
>
> [1] Lale et al. Regret minimization in partially observable linear quadratic control (2020).

---

### Official Review · Reviewer_rBWA · 2022-10-25

**Confidence:** 2
**Clarity, Quality, Novelty And Reproducibility:** The paper is clear and is of good qua…
**Correctness:** 3
**Technical Novelty And Significance:** 2
**Empirical Novelty And Significance:** Not applicable
**Recommendation:** 6

**Strength And Weaknesses:**

Strengths:

1. Sim-to-real transfer is an important problem in practice. Nonetheless, its theoretical analysis is lacking. The paper contributes to this less explored area.

2. The upper and lower bounds on the sim-to-real gap are novel, to my knowledge.

3. A detailed algorithm is provided to conduct sim-to-real transfer.

Some suggestions:

1. Sim-to-real transfer is important in many real-world applications. This problem is application driven. However, I am not sure if the upper and lower error bounds of the sim-to-real gap would offer practitioners any useful insight when conducting transfer learning. The author(s) might want to heavily revised the article to better discuss the insights you obtained that are useful to practical situations.

2. According to my understand, the paper borrows some techniques in minimax learning theory and seems to be an application of these techniques to sim-to-real transfer. The author(s) might want to discuss whether such an application is challenging and if the established results are sufficiently novel beyond the existing literature.

3. $\pi_{RT}$ is defined in a population level. It would be better if more details can be provided on how to obtain this oracle policy. In addition, the definition does not involve the sample size used for training. The theoretical results in Theorems 1 & 2 seem to be asymptotic. It also did not involve many other practically important parameters (e.g., the size of training data).

4. Algorithm 1 shall be highlighted in the paper instead of putting it in Section 4. The paper would benefit from a detailed discussion on the advances of the this algorithm beyond the existing literature.

5. The LQG model assumption is not very realistic in real applications. In the introduction, the author(s) mentioned that the partial observability was motivated by the OpenAI environments. However, do these environments satisfy the LQG model assumption?



**Summary Of The Paper:**

The paper considers sim-to-real transfer and approaches this problem from a theoretical perspective. It uses LQG to model the system dynamics and establishes the upper and lower bounds on the sim-to-real gap. A new algorithm is also developed for transfer learning.

**Summary Of The Review:**

The paper provides a theoretical analysis for sim-to-real transfer. I feel the paper needs substantial revision to better highlight its proposed algorithm. In addition, the theoretical analysis can also be strengthened.

---

> ### Author Response · Authors · 2022-11-13
> **Response to Reviewer rBWA**
>
> Thank you for your suggestions and comments.
>
> **Regarding the practical insights provided by the upper bound and lower bound**: We believe the upper
> bound and lower bound of the sim-to-real gap proved in our paper provide at least three practical insights, which
> are also discussed at the end of the introduction section. First, as many practical algorithms aim to find the robust
> adversarial training policy (see e.g., Pinto et al. (2017), Dennis et al. (2020)), we show that this policy is highly
> non-trivial with $\tilde{O}(\sqrt{H})$ sim-to-real gap. As a comparison, the sim-to-real gap of a trivial policy is $O(H)$. Second,
> we have computed the quantitative relationship between the sim-to-real gap of $\pi_{\mathrm{RT}}$ and the intrinsic complexity of
> the simulator class. This relationship implies the sim-to-real gap will be smaller for simpler simulator classes (i.e., simulator classes with smaller intrinsic complexity) under the realizability assumption. At last, we have proved that $\Omega(H)$ lower bound for the sim-to-real gap is unavoidable for any training policy, which implies that we cannot train a policy only in the simulation environments
> with perfect performance in any real-world task.
>
> **Regarding the challenges of our study and novelty of the technical results**: Our technical contributions
> have been summarized at the end of the introduction, which are mostly related to reinforcement learning instead
> of the minimax learning theory. The sim-to-real transfer in continuous domain with partial information is very
> challenging, as pointed out in the introduction of our paper (see the third paragraph of the introduction, and the
> paragraphs thereafter in the introduction). To resolve these challenges, we have proposed a novel reduction scheme,
> and a base algorithm LQG-VTR to accomplish the reduction.
>
> Our first technical contribution is a novel reduction argument, reducing the sim-to-real gap of $\pi_{\mathrm{RT}}$ to designing
> an algorithm with small regret bound in the infinite horizon average cost LQG problem. Then we turn to design
> such an algorithm, namely LQG-VTR, and prove its regret bound. The LQG-VTR algorithm is new. It uses an
> estimating procedure adopted by Lale et al. (2021) as a warm-up stage, then it follows optimistic planning and the
> value target regression proposed by Ayoub et al. (2020) to compute the optimal policy. To the best of our knowledge,
> the application of value target regression to the (partially observable) LQG problem is novel, and there are a number
> of challenges that need to be addressed in proving the regret bound of LQG-VTR (e.g., the boundedness of the
> belief states, the perturbation analysis of clipping the history, etc.). Moreover, we have also computed the intrinsic
> complexity of some simulator classes (see Appendix G), and proved that it scales only logarithmically with $H$ for
> any simulator class.
>
> **Regarding the implementation of the robust adversarial training oracle and the effect of practical
> parameters**: As mentioned in the review, the robust adversarial training policy $\pi_{\mathrm{RT}}$ is defined in the population
> level, which can be approximated accurately with a sufficient number of data during the simulation training phase.
> We study the sim-to-real gap of this population-level policy $\pi_{\mathrm{RT}}$ mainly because of two reasons:
> 1. It is the learning objective of robust adversarial training algorithms studied in many empirical works (see e.g.,
> Pinto et al. (2017), Dennis et al. (2020)). Therefore, a necessary condition to justify the robust adversarial
> training algorithms is to show that the sim-to-real gap of $\pi_{\mathrm{RT}}$ will be small.
> 2. It is cheap, fast, and safe to acquire a large number of training data in the simulation training phase since the
> policy is trained with simulators. Therefore, one can easily approximate $\pi_{\mathrm{RT}}$ in the simulation training phase
> given a large dataset.
>
> As for the implementation of the oracle that returns $\pi_{\mathrm{RT}}$, we have remarked in Section 2.4 that this oracle can
> be achieved by many previous algorithms in min-max optimal control and robust RL. We can also analyze how the
> approximation error relates to the training sample size following existing literature on online robust RL (see e.g.,
> Dong et al. (2022)) and offline robust RL (see e.g., Shi et al. (2022)). In practice, there are also many empirical
> algorithms that hope to learn the robust adversarial training policy $\pi_{\mathrm{RT}}$ (see e.g., Pinto et al. (2017), Dennis et al.
> (2020)). However, we want to emphasize that the approximation error in learning $\pi_{\mathrm{RT}}$ is not our focus in this paper.

---

> ### Author Response · Authors · 2022-11-13
> **Response to Reviewer rBWA**
>
> **Regarding highlighting the LQG-VTR algorithm**: We would like to note that the main idea of this paper
> is to show the robust adversarial training policy $\pi_{\mathrm{RT}}$ has a small sim-to-real gap. We achieve this goal via a reduction
> scheme. The algorithm LQG-VTR is proposed as part of the reduction. Although LQG-VTR is a novel algorithm for
> the regret minimization problem of infinite horizon average cost LQG systems, it is itself not a sim-to-real transfer
> algorithm.
>
> **Regarding the unrealistic assumption of linear quadratic Gaussian model in real applications**: The
> linear quadratic Gaussian control problem is one of the most fundamental optimal control problems in control theory.
> We believe the LQG model classes studied in this paper can be the first step towards understanding sim-to-real
> transfer in more complicated environments in the future, such as in POMDP model classes, or partially observable
> control systems with nonlinear dynamics.
>
> The LQG control has wide applications in physical engineering design, system analysis, image processing, etc.
> (see Wikipedia ”Linear–quadratic–Gaussian control”, ”Linear time-invariant system”). For example, Athans, M.
> (1971) reviewed how the optimal control of LQG can be applied to a general engineering control system design problem via
> a linearization technique. A wide class of physical processes can be regarded as such engineering design problems.
> The dexterous hand manipulation task (OpenAI. (2018)) may not inherit a linear dynamical transition in general.
> However, we can find an approximated solution with the linearization technique using LQG control.
>
> [1] Lale et al. Adaptive control and regret minimization in linear quadratic gaussian (lqg) setting (2021).
>
> [2] Ayoub et al. Model-based reinforcement learning with value-targeted regression (2020).
>
> [3] Dong et al. ”Online Policy Optimization for Robust MDP.” arXiv preprint arXiv:2209.13841 (2022).
>
> [4] Shi et al. ”Distributionally robust model-based offline reinforcement learning with near-optimal sample com-
> plexity.” arXiv preprint arXiv:2208.05767 (2022).
>
> [5] Pinto et al. Robust adversarial reinforcement learning (2017).
>
> [6] Dennis et al. Emergent complexity and zero-shot transfer via unsupervised environment design (2020).
>
> [7] Athans, M. The role and use of the stochastic linear-quadratic-Gaussian problem in control system design
> (1971).
>
> [8] OpenAI. Learning dexterous in-hand manipulation (2018).

---

### Official Review · Reviewer_YiUA · 2022-11-03

**Confidence:** 2
**Correctness:** 3
**Technical Novelty And Significance:** 3
**Empirical Novelty And Significance:** 3
**Recommendation:** 8

**Clarity, Quality, Novelty And Reproducibility:**

The paper is quite dense to read the main idea comes quite late after a lot of preliminaries (of which many are not being actively used in the main text). That being said, I appreciate the minimax formalization and the reduction of Lemma 3, which I had not seen before. It seems like an elegant approach.

**Strength And Weaknesses:**

Strengths:
* To the best of my understanding this is a sensible argument and there seems to be non-trivial novelty in this derivation.

Weaknesses:
* The paper is very dense and hard to unpack. Perhaps some of the technical details can be relegated to the appendix, giving more space for the main conceptual flow. Especially, for example, if the technical restrictions of the systems class are standard fare, they don't need to consume space in the main text given that they're not involved in any of the novel arguments.
* There are some writing nits, such as the fact that Gaussian is not capitalized in "linear quadratic gaussian" but is capitalized in "Gaussian distribution". As a proper name, Gaussian needs to be capitalized in every situation. There are also some tautologies, e.g.  "We follow
Chen et al. (2021) to present a formal formulation" (as opposed to an informal formulation). Generally, the writing would benefit from some clean-up.
* The preliminaries section seems to be very close to that of Lale's work. There are a number of common sentences. I do not believe this was done in bad faith, but the authors should consider rewriting this to avoid possible accusations of (auto)plagiarism.

**Summary Of The Paper:**

This paper studies the behavior of the sim-to-real gap for partially observed linear-Gaussian systems with quadratic cost. The authors formalize the sim-to-real gap as the finite-horizon minimax regret of a policy across a set of plausible simulators. The main argument is that the finite-horizon minimax regret can be bounded by a horizon-adjusted infinite-horizon regret up to an additive constant, hence it suffices to show that the infinite-horizon regret  minimization problem can be solved with bounded regret. The authors show this using an optimistic policy. There are, however, some number of technicalities to be taken care of along the way.

**Summary Of The Review:**

I would generally recommend acceptance of this paper due to the novelty of the formalization and its bound.

---

> ### Author Response · Authors · 2022-11-13
> **Response to Reviewer YiUA**
>
> Thank you for your comments and suggestions.
>
> **Regarding the hardness to unpack the paper and the involved technical arguments**: Thanks for
> pointing out this problem. We appreciate your suggestion and have revised our technical parts in the main text
> accordingly. Hopefully these updates help to better understand the main ideas.
>
> **Regarding moving the assumptions to the appendix**: We would like to keep the assumptions on the system
> class in the main paper because they are crucial to our results. For example, the history clipping scheme used in
> LQG-VTR, a core technique to establish our theory, is motivated by the strong stability assumption (Assumption
> 3).
>
> **Regarding the writing nits**: We have revised the paper accordingly. Thanks for pointing out this.
>
> **Regarding parts of the preliminary section being close to Lale et al. (2021)**: Thanks for the suggestions.
> We have revised the related parts.
>
> [1] Lale et al. Adaptive control and regret minimization in linear quadratic gaussian (lqg) setting (2021).

---

### Decision · Program_Chairs · 2023-01-20

**Decision:**

Accept: poster

**Justification For Why Not Higher Score:**

The writing of the paper needs to be improved to make it more accessible. Lack of intuition and insight makes it hard to appreciate the main techniques used in the paper.

**Justification For Why Not Lower Score:**

This is a good theory paper that studies the important sim-to-real problem in the non-trivial LQG setting.

**Metareview: Summary, Strengths And Weaknesses:**

This is a theoretical paper that studies sim-to-real in the linear quadratic Gaussian setting. The results are novel and the proofs contain non-trivial techniques. The writing and structure of the paper can be improved to make it more accessible. The reviewers have left comments that can help with this. I would like to thank the authors for making some changes to address the reviewers' comments. I would suggest to take the rest of their comments into account, especially those related to writing and presentation (making the work more accessible), in preparing the final draft of the paper.

**Note From Pc:**

if the above contains the word "oral" or "spotlight" please see: "oral" presentation means -> notable-top-5% and "spotlight" means -> notable-top-25%. As stated in our emails, we are disassociating presentation type from AC recommendations